# OBSERVATIONAL AUDITING OF PRIVACY

## ABSTRACT

Differential privacy (DP) auditing is essential for evaluating privacy guarantees in machine learning systems. Existing auditing methods, however, pose a significant challenge for large-scale systems since they require modifying the training dataset—for instance, by injecting out-of-distribution canaries or removing samples from training. Such interventions on the training data pipeline are resource-intensive and involve considerable engineering overhead. We introduce a novel observational auditing framework that leverages the inherent randomness of data distributions, enabling privacy evaluation without altering the original dataset. Our approach extends privacy auditing beyond traditional membership inference to protected attributes, with labels as a special case, addressing a key gap in existing techniques. We provide theoretical foundations for our method and perform experiments on Criteo and CIFAR-10 datasets that demonstrate its effectiveness in auditing label privacy guarantees. This work opens new avenues for practical privacy auditing in large-scale production environments.

## 1 INTRODUCTION

Differential privacy (DP) auditing has become an important tool for evaluating privacy guarantees in machine learning systems. Recent advances in auditing methods that require only a single run have made it feasible to evaluate privacy for large-scale models without prohibitive computational costs (Steinke et al., 2024; Mahloujifar et al., 2025b). However, most existing auditing approaches still require modifying the training dataset by injecting known entropy or canary data, which limits their applicability in industry-scale environments, where modifications to the training data pipeline require significant engineering overhead.

In this work, we propose a novel auditing methodology that eliminates the need for dataset modification. Our approach enables privacy evaluation using the natural nondeterminism present in the data distribution itself. We formalize and empirically validate this methodology in the setting of auditing Label DP, generalizable to privacy guarantees for any protected attribute. This capability addresses a significant gap in current auditing techniques, which have primarily focused on membership inference attacks (Shokri et al., 2017; Carlini et al., 2022) rather than attribute inference. In particular, existing methods for auditing Label DP either require adding out-of-distribution canaries to the training set (Malek et al., 2021) or apply only to a limited set of mechanisms (Busa-Fekete et al., 2024).

Observational privacy auditing cannot be done unconditionally, without making certain assumptions about the underlying distribution (Hernán & Robins, 2020, Chapter 3). Unlike anecdotal instances of privacy violations (Barbaro & Zeller Jr, 2006; Narayanan & Shmatikov, 2008; Carlini et al., 2021), auditing seeks to provide statistically valid measurements of memorization. In other words, the objective of privacy auditing is to establish *causality*—demonstrating that a model behaves in a certain way *because* it was trained on specific data. Traditionally, most rigorous membership inference attacks establish and measure causal effects through randomized control trials (RCTs), which require interference with the training data (Zhang et al., 2025). By framing auditing as a security game between two adversarial parties, our approach eliminates the need for training-time intervention.

Our key assumption is the availability of a distribution that approximates the ground truth data distribution. Concretely, the Label DP auditing mechanism relies on access to a proxy label-generating distribution. The proxy does not need to match the ground truth distribution, provided the adversary cannot distinguish between them (with reasonable computational resources). Under this assumption, the counterfactual examples generated by the challenger can be used to evaluate the attack on the

model's claimed Label DP guarantees *without training data intervention*. The attack can be made practical by using a model other than the target model as the proxy distribution. Further, in an incremental learning setting, earlier model checkpoints can be used as the proxy distribution, thus requiring no additional model training and minimal engineering overhead.

We demonstrate that our observational auditing framework provides accurate privacy bounds that match those obtained by interventional methods. Through a series of cryptography-inspired games we establish the theoretical foundations for auditing privacy without training data manipulation. By lowering the complexity of privacy auditing, our approach enables its application in a wider variety of contexts.

**Our approach and key contributions.** In the rest of this section we focus on the high-level intuition and concepts, deferring the formal treatment for later. The standard definition of differential privacy is formulated as a bound on the stability in distribution of a randomized algorithm in response to small changes in its input, such as inclusion or exclusion of a single user's data. Consequently, the auditing of differentially private implementations is typically presented as empirically testing these bounds by detecting changes in the output distribution on close inputs.

Definitions equivalent or closely related to differential privacy can be framed using the notion of a *simulator* whose role is to "create the illusion of stability" while protecting individual records in its input (Mironov et al., 2009; Mahloujifar et al., 2025a). In this approach, the privacy definition is operationalized as a game played between the simulator ("the defender") and the adversary ("the attacker"), whose probability of success under the rules of this game is constrained by the privacy parameters.

Our first contribution is incorporating the simulator *implementation* into the auditing procedure. Simulation-based definitions are standard (indeed foundational) in theoretical cryptography (Goldwasser & Micali, 1982; Goldwasser et al., 1985). The key distinction is that the cryptographic simulator is always a thought experiment, present in the proof but never implemented in practice (except for didactic purposes). In our approach, the auditor actually tests the simulator's performance vis-à-vis the distinguisher in order to derive privacy loss estimates.

The main advantage of simulation-based DP definitions is that they are inherently compatible with observational auditing. Whereas the counterfactuals of the standard DP formulation are produced by the same process as the real-world execution, the simulator can be implemented quite differently. The other advantage of simulation-based DP, as used in Mahloujifar et al. (2025a), is the extension of the add/remove DP framework (rather than replacement) to the privacy of partial records.

Auditing simulation-based differential privacy (DP) differs significantly from the standard approach. Instead of evaluating privacy by having a (fixed, domain-agnostic) challenger evaluate the attacker's success in breaching privacy guarantees, the process involves two competing algorithms: the simulator and the attacker. The simulator's goal is to generate outputs that closely resemble those of the real mechanism, but without access to certain masked data, thereby confusing the attacker. In this framework, the attacker may succeed for two main reasons: (1) the audited mechanism genuinely leaks private data, or (2) the simulator fails to effectively mimic the real mechanism. Consequently, the audit may underestimate privacy loss if the attacker is not sufficiently strong, or overestimate it if the simulator's outputs allow the attacker to easily distinguish between simulated and real data.

These trade-offs are not unique to our approach. In most cases, MIAs implicitly assume that the challenger can sample both members and non-members of the training set. Unless this capability is established before training the target model, typically through some form of intervention, the challenger's options are limited, and none are airtight (Zhang et al., 2025). By explicitly modeling the simulator as a participant in the privacy game, we allow the simulator's output to diverge from the true data distribution and can quantify the resulting slack in auditing guarantees.

A setting where this approach is especially effective is in auditing Label DP, which evaluates the extent to which classifiers memorize sensitive labels. In this context, the simulator's task is reduced to generating labels that closely match the true distribution. This task is not only fundamentally simpler than synthesizing complete data samples, but it also aligns directly with the objective of the target model. Thus, the architecture of the target model can be readily adapted for use by the simulator.

Our theoretical analysis builds upon the one-run auditing framework of Steinke et al. (2024) and Mahloujifar et al. (2025b). The novel part of the argument incorporates an additional pa-

rameter $\tau$ that controls the fidelity of the simulator into tail bounds on the adversary's advantage as a function of privacy parameters. We demonstrate that the audit remains accurate as long as we have a high quality estimator of the true data distribution.

To summarize, in this work we:

- Introduce a privacy auditing framework that does not interfere with model training. Instead, it runs a post-training game between a simulator that produces synthetic records (or record attributes) and an attacker that distinguishes between training and synthetic records.
- Generalize differential privacy auditing guarantees from Mahloujifar et al. (2025b) to account for the gap between the simulator and the true data distribution.
- Instantiate our framework for the case of auditing Label DP, giving an easy to implement attack in production settings, where labels are synthesized using earlier model checkpoints. We evaluate this attack empirically and show that it captures similar levels of memorization as interventional MIAs.

## 2 BACKGROUND AND PRIOR WORK

Differential Privacy (DP) introduced by Dwork et al. (2006) is a leading framework for providing rigorous privacy guarantees in statistical data analysis and machine learning. In its standard formulation, DP bounds the impact that any single individual's record has on the outcome of a computation by constraining how much the distribution of outputs may differ between neighboring datasets. The definition's strong theoretical guarantees, resilience to arbitrary auxiliary data, and compositional properties have driven its adoption in academic research and industry deployments (Fioretto & Hentenryck, 2025).

The general DP framework can be adapted to settings where only certain parts of a dataset are considered private. This paper focuses on Label DP, which has emerged as an important objective for privacy preserving machine learning, particularly in the domain of recommendation systems (Chaudhuri & Hsu, 2011; Ghazi et al., 2021; Malek et al., 2021; Wu et al., 2023). The following factors motivate Label DP as a uniquely valuable privacy concept:

- The label—representing the user's choice, expressed preference, or the outcome of an action—may be the only sensitive part of the record, with the rest being publicly available, static, non-sensitive data.
- In machine learning, labels are particularly vulnerable to memorization compared to other attributes since they most directly influence the loss function.
- In settings with mixed public/private features, instead of applying privacy-preserving techniques to sensitive features, one may exclude them from the model, potentially sacrificing some accuracy. In supervised learning, however, labels are indispensable—there is no analogous alternative to omitting sensitive features.

Complementing the strong worst-case guarantees of DP that bound the privacy loss from above on all inputs, privacy auditing empirically measures the privacy loss on concrete instances, providing a lower bound on DP's numerical parameters. Privacy auditing can be used for finding bugs in claimed implementations of DP algorithms (Ding et al., 2018), advancing understanding of complex DP mechanisms (Malek et al., 2021; Nasr et al., 2023; 2025), or guardrailing models in a production environment (Agrawal & Book, 2025).

Privacy auditing consists of two components: a privacy game between the challenger and the attacker, and an auditing analysis that translates the attacker's success into lower bounds on the $(\epsilon, \delta)$-DP guarantee (or other forms of DP). The privacy game is characterized by the capabilities and resources of the parties, and the attacker's goals, such as reconstruction, membership or attribute inference. Auditing analyses, typically used for membership inference, can be applied to any stochastic privacy game (Swanberg et al., 2025). We show theoretical results for our label inference attack building on Steinke et al. (2024) and Mahloujifar et al. (2025b).

Membership inference attacks (MIAs)—where an adversary uses model access and knowledge of the data distribution to determine whether a sample was part of training—have received significant

attention in the literature (Shokri et al., 2017; Yeom et al., 2018; Salem et al., 2019; Sablayrolles et al., 2019; Song & Shmatikov, 2019; Nasr et al., 2019; Leino & Fredrikson, 2020; Carlini et al., 2022; Ye et al., 2022; Zarifzadeh et al., 2024; Bertran et al., 2023). This attack category directly maps to the differential privacy guarantee where two neighboring datasets differ in the presence of one training sample.

A less studied category is attribute inference attacks (Yeom et al., 2018) where the adversary reconstructs a protected attribute given access to a partial record, of which label inference attacks are a special case (Malek et al., 2021; Busa-Fekete et al., 2024). A difficulty and common pitfall with such attacks is to properly account for the adversary's baseline success, achieved by exploiting knowledge of the data distribution and correlations between the public and protected attributes (Jayaraman & Evans, 2022). The label inference attack of Malek et al. (2021) uses canaries with random binary labels, which sets the adversary's baseline accuracy to $0.5$ and allows Label DP to be audited via standard MIA analyses.

Most auditing methods in the literature are "interventional," as their privacy game involves manipulating the training dataset: MIAs require excluding a subset of the data from training (Zanella-Beguelin et al., 2023; Steinke et al., 2024), whereas Malek et al. (2021) modifies the training labels. Training-data interventions severely restrict the applications of these auditing methods and may degrade model performance if too many samples are withheld or out-of-distribution canaries are injected. Instead, our label inference attack can run entirely post-training. MIAs can be stated as observational privacy games if the challenger is able to sample fresh samples from the distribution (Ye et al., 2022). However, obtaining new samples from the distribution (or from its close approximation), without affecting the training pipeline, remains an extremely challenging open problem (Maini et al., 2024; Duan et al., 2024; Meeus et al., 2025; Das et al., 2025).

Kazmi et al. (2024) propose an observational MIA that uses generative models such as generative adversarial networks (GANs) to obtain fresh samples. Differently from Kazmi et al. (2024), we propose a more general observational auditing framework (with observational MIA and LIA as special cases), and provide theoretical lower bounds on the privacy guarantees for these general attacks. A challenge with using synthetic samples is to account for how the shift between the generative model and the true distribution boosts the success of the attacker. Counterfactual labels are much easier to generate than entire samples (models are trained to do precisely this), and we can assume that an adversary with limited computation resources cannot distinguish between a label from the distribution and a label predicted from the best model the adversary can train.

The closest observational label inference attack to ours is the recent work of Busa-Fekete et al. (2024), which measures the label reconstruction advantage of the adversary with and without access to the model. This metric is not translated into a lower bound on $\epsilon$ in the differential privacy guarantee; in fact, such translation would be difficult because the adversary has a different baseline success (prior) for each sample. Additionally, this approach requires estimating probabilities of the mechanism's output given a particular label, limiting its applicability to simple mechanisms like randomized response and random label aggregation. In contrast, our label inference attack is applicable to all mechanisms. It can audit label privacy in a statistically valid manner because it sets up a game where the baseline accuracy of the adversary is $0.5$ for all samples.

## 3 PRELIMINARIES

We use calligraphic letters such as $\mathcal{X}, \mathcal{Y}, \mathcal{D}$ to denote sets and distributions. Capital letters such as $X, Y, D$ denote random variables and datasets, and lowercase letters denote their values. $\mathcal{X}^n$ is the set of all datasets of size $n$ with elements from $\mathcal{X}$, whereas $\mathcal{X}^*$ is the set of all finite-size data sets with elements from $\mathcal{X}$.

In this work, we audit the simulation-based definition of DP, introduced by Mahloujifar et al. (2025a). It generalizes the traditional add/remove (or "leave-one-out", or "zero-out") notion of DP to support a privacy unit that is a subset of the sample's attributes. The definition compares the distribution of a mechanism $M$ on dataset $D$ with that of a simulator that emulates the output of $M$ on $D$ without seeing the protected attributes of a record. This privacy definition fits nicely with our auditing framework, which is based on a proxy distribution (or simulator) that emulates training records. See Appendix A for further discussion of this definition.

**Definition 1** (Simulation-based privacy for protected attributes (Mahloujifar et al., 2025a)). *Let records $(x, y) \in \mathcal{X} \times \mathcal{Y}$ be such that $x$ is public or non-sensitive and thus need not be protected. We say that a randomized mechanism $M \colon (\mathcal{X} \times \mathcal{Y})^* \to \mathcal{Z}$ is $(\epsilon, \delta)$-SIM-DP with respect to a simulator $\mathrm{Sim} \colon (\mathcal{X} \times \mathcal{Y})^* \times \mathcal{X} \to \mathcal{Z}$ if for all datasets $D \in (\mathcal{X} \times \mathcal{Y})^*$, $(x, y) \in D$, and $D' = D \setminus \{(x, y)\}$, it holds*

$$M(D) \approx_{\epsilon, \delta} \mathrm{Sim}(D', x). \tag{1}$$

*For the more advanced notion of f-DP (Dong et al., 2020), a mechanism is f-SIM-DP with respect to $\mathrm{Sim}$ if $M(D) \approx_f \mathrm{Sim}(D', x)$. We also say $M$ is $(\epsilon, \delta)$-SIM-DP (resp. f-SIM-DP) if there exists a simulator $\mathrm{Sim}$ for which (1) (resp. (1)) holds.*

Our auditing guarantees are stated for a family of generic simulators that treat $M$ as a black-box. Such a simulator imputes the missing part $y$ of the record based on the public part $x$ and runs the original mechanism.

**Definition 2** (Imputation-based simulator). *For a mechanism $M \colon (\mathcal{X} \times \mathcal{Y})^* \to \mathcal{Z}$, a data distribution $\mathcal{D}$ supported on $\mathcal{X} \times \mathcal{Y}$, a dataset $D \in (\mathcal{X} \times \mathcal{Y})^*$ and public part of a record $x \in \mathcal{X}$, the imputation-based simulator $\mathrm{Sim}_{M, \mathcal{D}}$ is defined as*

$$\mathrm{Sim}_{M, \mathcal{D}}(D, x) \triangleq M(D \cup \{(x, y')\}), \quad \text{where } y' \sim \mathcal{D} \mid x.$$

Finally, we define empirical privacy auditing. A similar definition holds for $(\epsilon, \delta)$-SIM-DP.

**Definition 3** (Auditing simulation-based DP). *An audit procedure takes the description of a mechanism $M$, a trade-off function $f$, a simulator $\mathrm{Sim}$ and decides whether the mechanism satisfies f-SIM-DP with respect to $\mathrm{Sim}$. We define it as a two-step process.*

- Game: $\mathcal{M} \times \mathcal{S} \to \mathcal{O}$. *The auditor runs a potentially randomized experiment/game using the description of mechanism $M \in \mathcal{M}$ and the simulator $\mathrm{Sim}$. The auditor receives the game output $o \in \mathcal{O}$.*

- Evaluate: $\mathcal{O} \times \mathcal{F} \to \{0, 1\}$. *The output is $0$ if the auditor rejects the hypothesis that $M$ satisfies f-SIM-DP based on evidence $o$, and $1$ otherwise.*

## 4 OBSERVATIONAL VERSUS INTERVENTIONAL PRIVACY GAMES

An observational privacy game considers the training dataset as a given. In contrast, an interventional privacy game interferes with the training data pipeline and the eventual dataset used for training. In this section, we formalize our observational privacy auditing framework. To that end, we introduce a generic attack game, generalizing Swanberg et al. (2025), as Algorithm 1.

The privacy game occurs between two parties: a challenger and an adversary. A key difference from Swanberg et al. (2025) is that we split the challenger algorithm into two stages: sampling the training data ($G_{\mathsf{data}}$) and sampling the additional outputs provided to the adversary ($G_{\mathsf{hint}}$). With this, we can separate observational games from interventional ones. Further, we distinguish between the training dataset $D$ and additional game artifacts $S$, which are used by the challenger to set up a stochastic game (e.g., sampling random bits).

---

**Algorithm 1** Generic Attack Game (adapted from Swanberg et al. (2025))

---

**Input** Mechanism $M(\cdot)$, data distribution $\mathcal{D}$, distribution for game artifacts $\mathcal{D}_{\mathtt{prior}}$, adversary $A$
1: Sample training dataset $D = (x_1, \ldots, x_m)$ where $x_i \sim \mathcal{D}$.
2: Sample game artifacts $S \sim \mathcal{D}_{\mathtt{prior}}$.
3: Let $\overline{D} \leftarrow G_{\mathsf{data}}(D, S)$.  ▷ $G_{\mathsf{data}}$ determines the training dataset for $M$
4: Let $o_1 \leftarrow M(\overline{D})$.
5: Let $o_2 \leftarrow G_{\mathsf{hint}}(o_1, D, S)$.  ▷ $G_{\mathsf{hint}}$ determines additional input provided to the attacker $A$
6: Run attack $A(o_1, o_2)$ with access to $\mathcal{D}, \mathcal{D}_{\mathtt{prior}}, M$.
7: Measure adversary success with loss metric $\mathcal{L}(A(o_1, o_2), S)$.

---

The algorithm $G_{\mathsf{data}}$ determines the training dataset for $M$, obtained from a potential modification of the fixed training set $D$. For instance, in the one-run MIA (Steinke et al., 2024) the artifacts are

$S = (S_i \sim \text{Bernoulli}(0.5) : i \in [m])$, where $m$ is the number of canaries. The training dataset $\overline{D}$ is obtained from $D$ by including samples $x_i \in D$ where $S_i = 1$ and excluding samples where $S_i = 0$.

The role of $G_{\text{hint}}$ is to collect additional information the challenger provides to the adversary, based on game artifacts, training data, and the output of the trained model $M(\overline{D})$. In the one-run MIA, $o_2$ is the vector of targets $x_1, \ldots, x_m$ from the training set $D$.

**Definition 4** (Observational games). *We call a privacy game, as outlined in Algorithm 1, observational if $\overline{D} = D$ (and as a result $o_1 = M(D)$). That is, $M$ is trained on the original $D$, and the observations of the adversary consist of (1) output of $M(D)$ and (2) additional postprocessing of $D$, $M(D)$ according to the game artifacts $S$.*

**Observational/interventional MIA.** We have described how the (interventional) one-run MIA (Steinke et al., 2024) can be framed as Algorithm 1. We now present an observational one-run MIA (following Ye et al. (2022)).

Given data distribution $\mathcal{D}$, sample a sequence $S$ of game artifacts $S_i = (b_i, x_i') \sim \{0, 1\} \times \mathcal{D}$ for $i \in [m]$. The bits $b_i \sim \text{Bernoulli}(0.5)$, whereas $x_i$ is a fresh sample from the distribution (which is highly unlikely to be in $D$). Then, train model $M$ on $\overline{D} = D$. Let $o_2[i] = x_i$ if $b_i = 0$ and $o_2[i] = x_i'$ if $b_i = 1$ for $i \in [m]$. That is, the adversary receives either a training sample $x_i$ or a sample $x_i'$ from the distribution with probability 0.5. The adversary has to guess $b_i$, i.e., which of the samples it is seeing. This game is observational because the training dataset for $M$ remains unchanged.

From the adversary's perspective, the observational one-run MIA has the same distribution as the interventional MIA. For the challenger, the difference matters as the observational game does not alter the training pipeline. In this game, the source of counterfactual samples $x_i'$ can be a distribution that approximates $\mathcal{D}$ sufficiently well, as is the case with the attack of Kazmi et al. (2024) who use generative models to sample $x_i'$.

## 5 OBSERVATIONAL ATTRIBUTE INFERENCE

In this section, we describe our observational attribute inference attack. It allows privacy measurement with respect to any set of protected attributes, which can be the entire record (as in the observational MIA, Section 4) or just the label, for Label DP auditing. We provide theoretical results for obtaining empirical privacy lower bounds from our game. In particular, our analysis provides lower bounds on simulation-based DP in the add/remove privacy model (see Appendix A).

---

**Algorithm 2** Observational attribute inference in one run

**Input** Oracle access to a mechanism $M(\cdot)$, data distribution $\mathcal{D}$ and approximate distribution $\mathcal{D}'$ supported on $\mathcal{X} \times \mathcal{Y}$, attacker $A$

1: Let $D^0 = \left((x_1, y_1^0), \ldots, (x_m, y_m^0)\right)$, where $(x_i, y_i^0) \sim \mathcal{D}$ for $i \in [m]$.
2: Run mechanism $M$ on $D^0$.
3: Sample game artifacts $\left((b_1, y_1^1), \ldots, (b_m, y_m^1)\right)$ such that $(b_i, y_i^1) \sim \text{Bernoulli}(0.5) \times \mathcal{D}' \mid x_i$.
4: Construct a dataset $D^b = \left((x_1, y_1^{b_1}), \ldots, (x_m, y_m^{b_m})\right)$.
5: Run attack $A$ with input $M(D^0)$, $D^b$, and access to $\mathcal{D}, \mathcal{D}'$.
6: Reconstruct a vector of predictions $b' = (b_1', \ldots, b_m')$ which is supported on $\{0, 1, \perp\}^m$.
7: Count $c$, the number of correct guesses where $b_i' = b_i$, and $c'$, the total number of guesses where $b_i' \neq \perp$.   ▷ $\perp$ indicates abstention from guessing
8: **return** $(c, c')$.

---

Similar to prior auditing papers (Mahloujifar et al., 2025b; Steinke et al., 2024) the adversary can choose to abstain from guessing on samples where it is least confident, to boost its positive likelihood ratio. The observational game can use the entire dataset as canaries (i.e., $m = n$).

**Implementing the attack in practice, i.e., obtaining approximate distributions.** A key aspect in implementing the observational attribute inference attack is to produce the proxy distribution $\mathcal{D}'$ from which the counterfactual partial records $y_i^1$ are sampled. One option is to train an additional model $M'$ to predict the missing attribute(s). For label inference attacks, which are a special case

of Algorithm 2, we sample counterfactual label $y_i^i$ from Multinoulli($M'(x_i)$). In an online machine learning system, where the model trains incrementally as more recent data becomes available, one can use a prior model checkpoint as the model $M'$ (and run the attack on the newer data). This eliminates the need for training any additional models, making our proposed label inference attack very lightweight in terms of computational overhead and implementation complexity.

It is harder to operationalize Algorithm 2 for the case of MIA or attribute inference attacks, which require synthesizing entire samples or multiple attributes. In this case, diffusion models or generative adversarial models (GANs) can be used to generate the synthetic records, as in Kazmi et al. (2024). However, the architecture of the generative model might differ from the architecture of the target model, thus requiring training an additional model. This presents a tradeoff in implementation complexity between training an additional model versus modifying training data pipelines to exclude data. In contrast, the observational label inference attack uses prior model checkpoints and requires no additional training.

**Auditing guarantees under distribution shift**. We now establish theoretical auditing guarantees that translate the attacker's accuracy into a lower bound on the privacy parameters. Our bound depends on the total variation (TV) distance between $\mathcal{D} \mid x$ and $\mathcal{D}' \mid x$. Intuitively, the larger the distance between the two distributions, the weaker the lower bound we can obtain on the privacy guarantee.

**Theorem 5** (Auditing $f$-DP with distribution shift). *Let $M \colon (\mathcal{X}, \mathcal{Y})^* \to \mathcal{Z}$ be a mechanism, $\mathcal{D}$ the data distribution, $\mathcal{D}'$ an approximate distribution, and $\mathrm{Sim}_{M,\mathcal{D}'}$ the imputation-based simulator (Definition 2). Let $C = \sum_{i \in [m]} \mathbf{1}[b_i' = b_i]$ be the total number of correct answers from the one-run observational attribute inference attack (Algorithm 2) for an adversary that makes $c'$ guesses. Let $\mathrm{TV}(\mathcal{D}|x, \mathcal{D}'|x) \leq \tau$ for all $x$ in the dataset $D$ and define $g \colon [0,1] \to [0,1]$ such that*

$$g(s) = f(\min(1, s + \tau)). \tag{2}$$

*If $M$ is $(\epsilon, \delta)$-SIM-DP with respect to $\mathrm{Sim}_{M,\mathcal{D}'}$ and the auditing Algorithm 3 returns False on $(c', c, M, g, \gamma)$, then $\Pr[C \geq c] \leq \gamma$.*

**Accounting for the distribution shift in practice.** Theorem 5 requires that the bound $\tau$ be known to the challenger, which in practice can be hard to estimate. Not accounting for $\tau$ properly leads to an *overestimation* of the privacy leakage. Nonetheless, for the case of Label DP auditing we recommend using Theorem 5 with $\tau = 0$ if the challenger is training the best available model for the prediction task. To elaborate, Theorem 5 can also be stated in terms of the adversary's ability to distinguish between $\mathcal{D}$ and $\mathcal{D}'$ given its knowledge and resource constraints. The value $\tau$ in Theorem 5 is an upper bound on the adversary's a priori (i.e., before having access to the target model) success probability in distinguishing whether a sample $(x, y^b)$ is from $\mathcal{D}$ or $\mathcal{D}'$. In the case of Label DP auditing, if $M'$ denotes the best classifier on $\mathcal{D}$ that the adversary can access, we may reasonably assume that the adversary is unable to distinguish the true conditional label distribution $y \mid x$ from the distribution Multinoulli($M'(x)$).

In Fig. 4 (Appendix C) we empirically show the impact of $\tau$ on the measured privacy loss. The larger $\tau$, the lower the measured privacy loss, due to the additive $\tau$ term in Eq. 2. While the the additive dependence can significantly degrade the measured empirical privacy loss, this can be improved by assuming stronger bounds on the distribution divergence (e.g., using Kullback-Leibler divergence as opposed to TV distance).

# 6 EXPERIMENTS ON AUDITING LABEL DP

We validate our theoretical framework with experiments on two representative datasets: CIFAR-10 (Krizhevsky, 2009) for image classification and Criteo (Criteo AI Lab, 2015) for tabular data with sensitive labels. For each dataset, we train classifiers using standard Label DP mechanisms and empirically evaluate the privacy guarantees using our audit procedure. Appendix E shows similar results for a third large text dataset. We also evaluate our Label DP auditing technique for Randomized Response (Warner, 1965) on synthetic data.

## 6.1 LABEL DP LEARNING ALGORITHMS

The earliest approach to achieving Label DP is the Randomized Response (RR) mechanism (Warner, 1965). In RR, each training label is randomly replaced according to a fixed probability distribution

before being shared with the learning algorithm. This randomization helps protect the privacy of individual labels. We briefly review several recent Label DP mechanisms that improve on RR.

**Label Private One-Stage Training (LP-1ST, Ghazi et al. (2021))** Instead of using a fixed distribution as in RR, LP-1ST samples each training label $y_i$ from a learned prior distribution $P(y \mid X_i)$. The prior can be estimated by observing the top-$k$ predictions from a pretrained model (either in domain or out-of-domain), restricting RR to the most probable labels. Alternatively, the training can be split into multiple stages, where an earlier model provides the prior for the next (LP-MST).

**Private Aggregation of Teacher Ensembles with FixMatch (PATE-FM, Malek et al. (2021))** PATE-FM combines the FixMatch semi-supervised learning algorithm (Sohn et al., 2020) with private aggregation. Multiple teacher models are trained, each using all unlabeled examples and disjoint subsets of the labeled data. The predictions from these teachers are then aggregated in a differentially private manner using the PATE framework (Papernot et al., 2017) to train a student model.

**Additive Laplace with Iterative Bayesian Inference (ALIBI, Malek et al. (2021))** ALIBI achieves Label DP by adding Laplace noise to the one-hot encoded labels (Ghosh et al., 2012), making the released labels differentially private. The model is trained on soft labels computed from noisy observations via Bayesian inference.

## 6.2 ATTACK IMPLEMENTATION AND ADVERSARIAL STRATEGY

A key ingredient in implementing our label inference attack is generating the reconstructed label $y^1$ given features $x$ (see Algorithm 2). We generate $y^1$ from the predictions of a reference model $M'$, trained on separate data (but from approximately the same distribution) as the target model $M$. Specifically, $y^1 \sim \mathsf{Multinoulli}(M'(x))$, where $M'(x)$ are the predictions of $M'$ on $x$ for each class.

For the Criteo dataset, where data is collected over 28 consecutive days, we train $M'$ on Day 0 data and the target model on Day 1 data. While there may be some distribution shift between Day 0 and Day 1 data, we assume this is small, i.e., ($\tau$ in Theorem 5 is 0) and that the adversary cannot distinguish between the true labels $y^0$ and the reconstructed labels $y^1$ without access to the model. For CIFAR-10 experiments we randomly split the training data ($n = 50$K samples) into two. We train $M'$ on the first half, and the target model $M$ on the second half.

We run the attack on $m = 200$K canaries for Criteo and $m = 10$K canaries for CIFAR-10. One of the benefits of our observational attack is that it can be run with as many canaries as the size of the training set, eliminating a prior tradeoff for interventional attacks, where using more canaries gives tighter confidence intervals but decreases the size of the available training set. To use the same number of canaries as for MIA experiments, the number of canaries is set to the size of the test sets.

The adversary obtains its guesses by computing per-example scores. Let $x$ be the features and $y^b \in \{y^0, y^1\}$ be the label received by the adversary. The adversary computes a score that correlates with whether $y^b$ is the reconstructed or the training label. The score consists of two components. The first component $s_1(x, y^b)$ is the difference in probabilities that $y^b$ came from the training set versus the reconstructed distribution:

$$s_1(x, y^b) \triangleq \Pr[y^0 = y^b \mid M(x)] - \Pr[y^1 = y^b \mid M'(x)]$$
$$= M(x)[y^b] - M'(x)[y^b],$$

where $M(x)[y^b]$ is the prediction of $M$ on $x$ for class $y^b$. Since the adversary's performance is measured at the tails of the score distribution, the adversary prefers to guess on samples where $\Pr[y^0 \neq y^1]$ is high. Thus the second component of the score is defined as

$$s_2(x, y^b) \triangleq \Pr[y^0 \neq y^1] = 1 - M'(x)[y^b].$$

The final score combines the two components as $s(x, y^b) \triangleq s_1(x, y^b) \cdot s_2(x, y^b)^t$ with a hyperparameter $t \geq 0$ that allows for weighting the two components separately. We use $t = 2$, as $t > 1$ gives tight lower bounds for RR. The adversary guesses on $c'\%$ of the samples with the highest absolute scores. We sweep $c' \in \{1, 2, \ldots, 100\}$ and report the highest $\epsilon$ achieved at 95% confidence, averaged over 100 repetitions of the game (resamplings of counterfactual labels). In Appendix E, we show how we obtain similar results for a fixed $c'$.

Table 1: CIFAR-10. Auditing Label DP algorithms under different $\epsilon$ with $\delta = 10^{-5}$.

| Label DP Algorithm | CIFAR-10 | | |
| --- | --- | --- | --- |
| | $\epsilon = \infty$ | $\epsilon = 10.0$ | $\epsilon = 1.0$ |
| LP-1ST | $2.13 \pm .22$ | $2.02 \pm .33$ | $0.43 \pm .05$ |
| LP-1ST (out-of-domain prior) | $2.26 \pm .22$ | $1.86 \pm .25$ | $0.90 \pm .07$ |
| PATE-FM | $2.42 \pm .32$ | $2.22 \pm .24$ | $0.79 \pm .09$ |
| ALIBI | $2.53 \pm .33$ | $2.18 \pm .27$ | $0.67 \pm .07$ |

Table 2: Criteo. Auditing Label DP algorithms for different $\epsilon$ with $\delta = 10^{-5}$. As in Wu et al. (2023), for LP-1ST (domain prior) at $\epsilon \in \{0.1, 1, 2\}$ the training process did not produce meaningful outcomes.

| Label DP Algorithm | $\epsilon = \infty$ | $\epsilon = 8$ | $\epsilon = 4$ | $\epsilon = 2$ | $\epsilon = 1$ | $\epsilon = 0.1$ |
| --- | --- | --- | --- | --- | --- | --- |
| LP-1ST | $1.37 \pm .14$ | $1.29 \pm .12$ | $1.22 \pm .17$ | $0.59 \pm .05$ | $0.34 \pm .02$ | $0.06 \pm .01$ |
| LP-1ST (domain prior) | $1.47 \pm .22$ | $1.37 \pm .10$ | $1.28 \pm .21$ | — | — | — |
| LP-1ST (noise correction) | $1.31 \pm .12$ | $1.26 \pm .11$ | $1.04 \pm .16$ | $0.52 \pm .07$ | $0.40 \pm .08$ | $0.06 \pm .01$ |
| LP-2ST | $1.52 \pm .14$ | $1.46 \pm .11$ | $1.24 \pm .15$ | $0.75 \pm .05$ | $0.61 \pm .07$ | $0.06 \pm .01$ |
| PATE | $1.60 \pm .15$ | $1.48 \pm .14$ | $1.28 \pm .12$ | $0.71 \pm .07$ | $0.59 \pm .05$ | $0.06 \pm .01$ |

## 6.3 CIFAR-10 EXPERIMENTS

For CIFAR-10, we treat the image classes as sensitive labels. We train standard convolutional neural networks with varying privacy budgets $\epsilon \in \{1, 10, \infty\}$ (see model accuracy in Table 4, Appendix D). We then audit Label DP using our observational game and report results in Table 1.

## 6.4 CRITEO EXPERIMENTS

The Criteo dataset contains user click-through data with demographic information encoded as 13 numerical features and 26 categorical features. A binary label indicates whether the user clicked on the ad. The distribution of the labels is highly imbalanced, with only 3% of positives (clicks). The overall dataset contains over 4 billion click log data points over a period of 24 days. We followed the same setup in Wu et al. (2023) where 1 million data points are selected for each day. We divide the data into 80% for training, 4% for validation, and 16% for testing. Model performance is evaluated using the log-loss metric on the test set.

We train gradient boosting decision trees with the CatBoost library (Prokhorenkova et al., 2018) with varying privacy budgets $\epsilon \in \{0.1, 1, 2, 4, 8, \infty\}$ and evaluate label privacy using observational privacy auditing (Table 2). Table 5 (Appendix D) shows model performance under different Label DP algorithms and values $\epsilon$.

## 6.5 COMPARISON WITH EXISTING METHODS

We compare our observational auditing approach against traditional canary-based methods to demonstrate the effectiveness and practicality of our framework. More specifically, we evaluate against the lightweight *difficulty calibration* MIA in Watson et al. (2022), where membership scores are adjusted to the difficulty of correctly classifying the target sample. For each canary datapoint, we set the calibrated membership scores as the difference in the loss between the target model and the reference model $M'$. Fig. 1 shows how our method achieves similar auditing results when compared to MIA on the CIFAR-10 and Criteo datasets.

We leave to future work a comparison with more computationally intensive methods, such as Zarifzadeh et al. (2024), which require training multiple auxiliary (shadow) models to achieve state-of-the-art attack performances. However, we compare our approach to that of Zarifzadeh et al. (2024) for the special case of one single reference model in Appendix E.

**Gap between theoretical and empirical epsilon.** Fig. 1 shows a large gap between the theoretical and empirical bounds on $\epsilon$, especially for larger values of $\epsilon$. This gap is due to several factors that have been previously explored in the literature: (1) the attacks are black-box, i.e., they do not have access to intermediate outputs of the training procedure such as gradients (Nasr et al., 2021), (2) the theoretical $\epsilon$ is an upper bound that is not necessarily tight, (3) $f$-DP auditing is not always tight, especially at higher values of $\epsilon$ (Mahloujifar et al., 2025b).

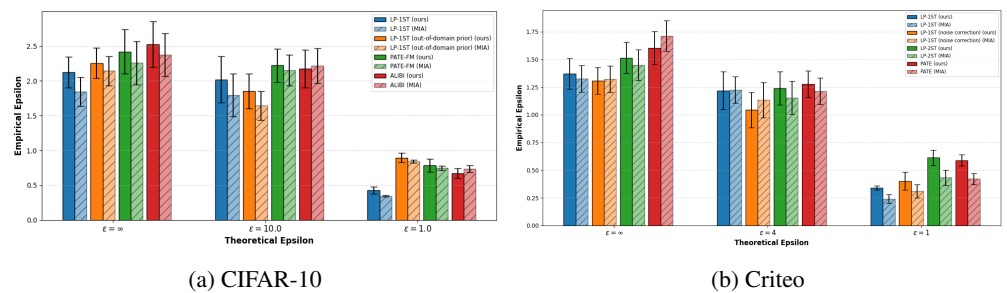

(a) CIFAR-10  (b) Criteo

Figure 1: Comparison with MIA for different Label DP Algorithms on CIFAR-10 and Criteo datasets. The error bar represents the standard deviation across 100 different repetitions.

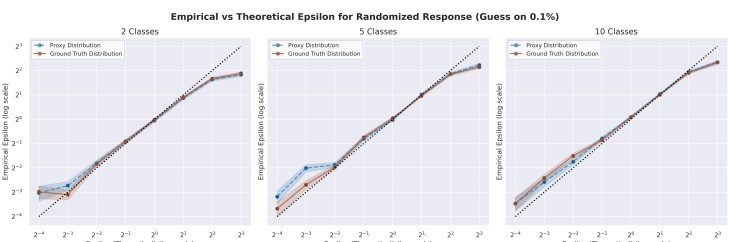

Figure 2: Auditing randomized response when the adversary guesses on 0.1% of samples. The counterfactual labels are generated either from the ground-truth distribution or sampled according to the output of a logistic regression model.

## 6.6 SYNTHETIC DATA AND AUDITING RANDOMIZED RESPONSE

We empirically demonstrate the tightness of our Label DP auditing algorithm for Randomized Response using synthetic data. The distribution consists of $k$ balanced classes, each generated from a 5-dimensional Gaussian with the same covariance but a shifted mean. We generate $n = 10^6$ samples and vary $k \in \{2, 5, 10\}$. Counterfactual labels $y^1$ are generated using either the true distribution or the predictions of a logistic regression model. Figure 2 shows empirical epsilon lower bounds at 95% confidence when the adversary makes 0.1% non-abstaining guesses. We obtain tight lower bounds for $\epsilon \in [1, 4]$. At lower epsilons, the audit overestimates privacy loss due to a higher variance induced by a small number of guesses. Using more guesses at lower epsilon fixes the issue (Appendix C).

## 7 DISCUSSION

In this paper, we establish a framework for auditing privacy without any intervention during the training process. This enables a principled privacy evaluation in settings where the training process is outside the control of the privacy auditor. This may sound counterintuitive as privacy auditing is a form of causal analysis. However, our method can provide provable guarantees on auditing performance under certain assumptions about the data distribution. We envision that our framework will broaden the scope of privacy auditing applications, as it does not require any supervision of the training process and can be conducted by third parties.

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

# A SIMULATION-BASED DIFFERENTIAL PRIVACY

In this section, we motivate and formally define the simulation-based notion of differential privacy. We recall the standard definition of differential privacy, which we are going to extend and adapt to our setting.

**Definition 6** (Differential privacy). *A randomized mechanism $M\colon \mathcal{X}^* \to \mathcal{Z}$ satisfies $(\epsilon, \delta)$-differential privacy if for all pairs of neighboring datasets $D, D' \subset \mathcal{X}^*$ and $E \subset \mathcal{Z}$ it holds:*
$$\Pr[M(D) \in E] \leq e^\epsilon \Pr[M(D') \in E] + \delta.$$

The definition depends on the notion of *neighboring datasets*, which is a symmetric binary relation on $\mathcal{X}^*$ denoted as $D \sim D'$. Choosing the neighboring relationship is an important part of mechanism design and has direct implications on the type of privacy guarantee. See Table 3 for some common definitions of neighboring datasets and the resulting DP notions.

Differentially private mechanisms are often applied to datasets containing records with both public and private components. Consider records of the form $(x, y) \in \mathcal{X} \times \mathcal{Y}$, where $x$ represents the public (non-sensitive) data and $y$ the sensitive attributes that require protection. Differential privacy for this setting can be defined by letting $D \sim D'$ if they differ only in the sensitive attributes of a single record: that is, $D$ and $D'$ are identical except for one record being $(x, y)$ in $D$ and $(x, y')$ in $D'$. This is a generalization of the original definition of Dwork et al. (2006), which modeled $D$ as an indexed vector. In current terminology, this is the "replacement" model of differential privacy: the sensitive portion of a record is *replaced* with a different value. Semantically, this guarantees that an adversary observing $M$'s output cannot distinguish between two possible private values $y$ and $y'$ of a user any better than without $M$, within an $(\epsilon, \delta)$-slack.

The alternative to the replacement notion is add/remove, which stipulates that $M$'s output on inputs with and without the user are $(\epsilon, \delta)$-indistinguishable. In addition to protecting the user's data, this model also hides the user's membership status and the size of the dataset. DP in the add/remove model implies DP in the replacement model (via the two-step hybrid, with looser parameters) but not vice versa.

We target the add/remove model of differential privacy. For datasets containing both public and private attributes, one way to define this model is by introducing a class of records where the sensitive parts are removed. In this formulation, two datasets $D$ and $D'$ are neighboring if they differ only in the pairs $(x, y)$ and $(x, \bot)$. We propose an equivalent definition that is more explicit, as it introduces the notion of a *simulator*:

**Definition** (more formal version of Definition 1, Simulation-based privacy for protected attributes). *Let records $(x, y) \in \mathcal{X} \times \mathcal{Y}$ be such that $x$ is public or non-sensitive and thus need not be protected. We say that a randomized mechanism $M\colon (\mathcal{X} \times \mathcal{Y})^* \to \mathcal{Z}$ is $(\epsilon, \delta)$-SIM-DP with respect to a simulator $\mathrm{Sim}\colon (\mathcal{X} \times \mathcal{Y})^* \times \mathcal{X} \to \mathcal{Z}$ if for all datasets $D \in (\mathcal{X} \times \mathcal{Y})^*$, $(x, y) \in D$, $D' = D \setminus \{(x, y)\}$, and $E \subset \mathcal{Z}$ it holds*
$$\Pr[M(D) \in E] \leq e^\epsilon \Pr[\mathrm{Sim}(D', x) \in E] + \delta.$$
*For the more advanced $f$-differential privacy notion, we call a mechanism $f$-SIM-DP with respect to $\mathrm{Sim}$ if*
$$\Pr[M(D) \in E] \leq f(\Pr[\mathrm{Sim}(D', x) \in E]).$$

Semantically, $(\epsilon, \delta)$-SIM-DP means that anything that can be inferred about $(x, y)$ from $M(D)$ could also be inferred without ever exposing $y$ to $M$, within the standard $(\epsilon, \delta)$-bounds.

Whenever we say that a mechanism satisfies $f$-(SIM)-DP, we implicitly imply that $f$ is a valid trade-off function. That is, $f$ is defined on the domain $[0, 1]$ and has a range of $[0, 1]$. Moreover, $f$ is decreasing and convex with $f(x) \leq 1 - x$ for all $x \in [0, 1]$. This is without loss of generality. That is, if a mechanism is $f$-(SIM)-DP for an arbitrary function $f\colon [0, 1] \to [0, 1]$, then it is also $f'$-(SIM)-DP for a valid trade-off function $f'$ with $f'(x) \leq f(x)$ for all $x \in [0, 1]$ (See Proposition 2.2 in Dong et al. (2020)).

Note that $f$-DP generalizes $(\epsilon, \delta)$-DP by allowing a more complex relation between the probability distributions of $M(D)$ and $M(D')$. The following proposition shows how one can express $(\epsilon, \delta)$-DP as an instantiation of $f$-DP. Its analogue also holds for $\mathrm{Sim}$-DP.

**Proposition 7** (Dong et al. (2020)). *A mechanism is $(\epsilon, \delta)$-DP if it is $f$-DP for $f$ satisfying $1 - f(x) = e^\epsilon \cdot x + \delta$.*

| DP definition | Difference between $D \sim D'$ |
|---|:---:|
| Replacement, DP | $x, x'$ |
| Add/remove, DP | $x$ |
| Replacement, Label DP | $(x, y), (x, y')$ |
| Add/remove, Label DP | $(x, y), (x, \bot)$ |

Table 3: The difference between two neighboring datasets $D, D'$ under various DP definitions.

# B    AUDITING $f$-DIFFERENTIAL PRIVACY

In this section, we prove Theorem 5 which generalizes the guarantees of Mahloujifar et al. (2025b) for auditing $f$-SIM-DP, allowing for the case when we sample the counterfactual attributes from a proxy distribution $\mathcal{D}'$ different from $\mathcal{D}$.

We first describe how the accuracy of the adversary from the observational attribute inference attack can be translated into a lower bound on $f$-SIM-DP.

**Definition 8** (Obtaining empirical epsilon from $f$-SIM-DP auditing). *Let* (Game, Evaluate) *be an audit procedure. The empirical privacy of a mechanism $M$ for a family $F$ of trade-off functions and a simulator* Sim *is the random variable distributed according to the output of the following process:*

*1: Obtain observation $o \leftarrow$ Game$(M, \mathrm{Sim})$.*
*2: Construct $F_o = \mathrm{maximal}\{f \in F \colon \mathrm{Evaluate}(o, f) = 1\}$, where the partial order on $F$ is defined as $f \prec g$ iff $f(x) \leq g(x)$ for all $x \in [0, 1]$.*
*3: Compute*

$$\epsilon(\delta) = \min_{f \in F_o} \max_{x \in [0,1]} \log\left(\frac{1 - f(x) - \delta}{x}\right).$$

The empirical lower bound $\epsilon(\delta)$ is a random variable since it is a function of the output $o$ of a randomized process Game. The point estimate $\epsilon(\delta)$ is the lowest $\epsilon$ given $\delta$ guaranteed by an $f$-SIM-DP *not* rejected by the auditing procedure.

In Algorithm 3 we show how to audit a particular trade-off function $f$ given the number of non-abstaining guesses $c'$ and the number of correct guesses $c$. (This is the Evaluate function in Step 2 of Definition 8). The choice of a family of trade-off functions $F$ in Definition 8 should be based on the expectations of the true privacy curve. For example, if one expects the privacy curve of a mechanism to be similar to that of a Gaussian mechanism, then one would choose the set of all trade-off functions imposed by a Gaussian mechanism as the family. This is the choice we use in our experiments. The overall auditing procedure executes Algorithm 3 for each function $f$ in $F$ in increasing order of privacy strength and reports the strongest privacy guarantee accepted by Algorithm 3.

Finally, we re-state and prove Theorem 5.

**Theorem 9** (Restated Theorem 5, auditing $f$-DP with distribution shift). *Let $M \colon (\mathcal{X}, \mathcal{Y})^* \to \mathcal{Z}$ be a mechanism, $\mathcal{D}$ the data distribution, $\mathcal{D}'$ an approximate distribution, and $\mathrm{Sim}_{M,\mathcal{D}'}$ the imputation-based simulator (Definition 2). Let $C = \sum_{i \in [m]} \mathbf{1}[b'_i = b_i]$ be the total number of correct answers from the one-run observational attribute inference attack (Algorithm 2) for an adversary that makes $c'$ guesses. Let $\mathrm{TV}(\mathcal{D}|x, \mathcal{D}'|x) \leq \tau$ for all $x$ in the dataset $D$ and define $g \colon [0,1] \to [0,1]$ such that $g(s) = f(\min(1, s + \tau))$. If $M$ is $(\epsilon, \delta)$-SIM-DP with respect to $\mathrm{Sim}_{M,\mathcal{D}'}$ and Algorithm 3 returns False on $(c', c, M, g, \gamma)$, then $\Pr[C \geq c] \leq \gamma$.*

*Proof.* The proof follows similarly to the proof of Theorem 3.2 in Mahloujifar et al. (2025b). We first need to prove a similar Lemma to that of their Lemma A.1 that is adapted to our setting of simulation based differential privacy with distribution shift.

**Lemma 10.** *Let $M \colon (\mathcal{X} \times \mathcal{Y})^* \to \mathcal{Z}$ be a mechanism, $\mathcal{D}$ a distribution on $\mathcal{X} \times \mathcal{Y}$ and $\mathrm{Sim}_{M,\mathcal{D}'}$ the imputation-based simulator (Definition 2). Assume $\mathrm{TV}(\mathcal{D}|x, \mathcal{D}'|x) \leq \tau$ for all $x$ in the dataset $D$. If $M$ is $f$-SIM-DP with respect to $\mathrm{Sim}_{M,\mathcal{D}'}$, then for any attack algorithm $A$ and event $E \subset \mathcal{Z}$ we have*

$$f''_\tau(\Pr[M(D) \in E]) \leq \Pr[M(D) \in E \ \& \ b_1 = b'_1] \leq f'_\tau(\Pr[M(D) \in E]),$$

**Algorithm 3** Iteratively deciding an upper bound probability of making more than $c$ correct guesses (Mahloujifar et al., 2025b)

---

**Input** description of trade-off function $f$, number of guesses $c'$, number of correct guesses $c$, number of samples $m$, probability threshold $\gamma$ (default is $\gamma = 0.05$).

1: $\forall 0 \leq i \leq c$ set $h[i] = 0$, and $r[i] = 0$.
2: Set $r[c] = \gamma \cdot \frac{c}{m}$.
3: Set $h[c] = \gamma \cdot \frac{c'-c}{m}$.
4: **for** $i \in [c-1, \ldots, 0]$ **do**
5:     $h[i] = \bar{f}^{-1}\big(r[i+1]\big)$   $\triangleright \overline{f}(x) \triangleq 1 - f(x)$
6:     $r[i] = r[i+1] + \frac{i}{c'-i} \cdot \big(h[i] - h[i+1]\big)$.
7: **end for**
8: **if** $r[0] + h[0] \geq \frac{c'}{m}$ **then**
9:     Return False   $\triangleright$ Probability of $c$ correct guesses (out of $c'$) is less than $\gamma$
10: **else**
11:     Return True   $\triangleright$ Probability of having $c$ correct guesses (out of $c'$) could be more than $\gamma$
12: **end if**

---

*where*

$$f'_\tau(s) \triangleq \sup\{t \in [0,s]; t + f(s-t+\tau) \leq 1\} \quad and \quad f''_\tau(s) \triangleq \inf\{t \in [0,1]; f(t)+s-t \leq 1-\tau\}.$$

*Proof.* Fix a sample $(x, y^0)$ with counterfactual attributes $y^1$. Let $D$ denote the dataset containing $(x, y^0)$ and let $D'$ be a dataset obtained from $D$ by replacing $y^0$ with $y^1$. We assume $b'_1$ is a deterministic function of $M(D)$ and $y^{b_1}$ (i.e., the attack is deterministic). Let $p \triangleq \Pr[M(D) \in E \ \& \ b_1 = b'_1]$ and $q \triangleq \Pr[M(D) \in E]$. We have

$$
\begin{aligned}
p &= \Pr[M(D) \in E \ \& \ b_1 = b'_1] \\
&= \Pr[M(D) \in E \ \& \ b_1 = 1 \ \& \ b'_1 = 1] \\
&\quad + \Pr[M(D) \in E \ \& \ b_1 = 0 \ \& \ b'_1 = 0] \\
&= \mathop{\mathrm{E}}_{y^1,y^0,b_1,\theta \sim M(D)}[I(\theta \in E \ \& \ b_1 = 1 \ \& \ b'_1(\theta, y^1) = 1)] \\
&\quad + \mathop{\mathrm{E}}_{y^1,y^0,b_1,\theta \sim M(D)}[I(\theta \in E \ \& \ b_1 = 0 \ \& \ b'_1(\theta, y^0) = 0)] \\
&= 0.5 \cdot \mathop{\mathrm{E}}_{y^1,y^0,b_1,\theta \sim M(D)}[I(\theta \in E \ \& \ b'_1(\theta, y^1) = 1) \mid b_1 = 1] \\
&\quad + 0.5 \cdot \mathop{\mathrm{E}}_{y^1,y^0,b_1,\theta \sim M(D)}[I(\theta \in E \ \& \ b'_1(\theta, y^0) = 0) \mid b_1 = 0] \\
&= 0.5 \cdot \mathop{\mathrm{E}}_{y^1,y^0,b_1,\theta \sim M(D),\theta' \sim M(D')}[I(\theta \in E \ \& \ b'_1(\theta, y^1) = 1) \mid b_1 = 1] \\
&\quad + 0.5 \cdot \mathop{\mathrm{E}}_{y^1,y^0,b_1,\theta \sim M(D),\theta' \sim M(D')}[I(\theta \in E \ \& \ b'_1(\theta, y^0) = 0) \mid b_1 = 0] \\
&\leq 0.5 \cdot \Big(1 - f\big(\mathop{\mathrm{E}}_{y^1,y^0,b_1,\theta \sim M(D),\theta' \sim M(D')}[I(\theta' \in E \ \& \ b'_1(\theta', y^1) = 1) \mid b_1 = 1]\big)\Big) \\
&\quad + 0.5 \cdot \Big(1 - f\big(\mathop{\mathrm{E}}_{y^1,y^0,b_1,\theta \sim M(D),\theta' \sim M(D')}[I(\theta' \in E \ \& \ b'_1(\theta', y^0) = 0) \mid b_1 = 0]\big)\Big) \\
&\leq 1 - f\big(\mathop{\mathrm{E}}_{y^1,y^0,b_1,\theta \sim M(D),\theta' \sim M(D')}[I(\theta' \in E \text{ and } b_1 \neq b'_1(\theta', y_1^{1-b_1})]\big) \quad \text{(By convexity of } f.)
\end{aligned}
$$

Now since, $\mathrm{TV}(\theta', \theta) \leq \tau$, we have $\Pr[[\theta' \in E] \leq \Pr[\theta \in E] + \tau$ for all $E$. this, by the fact that $f$ is decreasing implies,

$$p \leq 1 - f(q - p + \tau).$$

Therefore, $p + f(q - p + \tau) \leq 1$ and $p \leq f'_\tau(q)$. Similarly, for the other side we repeat the argument up until the last 3 steps. That is

$$q = 0.5 \cdot \underset{y^1, y^0, b_1, \theta \sim M(D), \theta' \sim M(D')}{\mathrm{E}} [I(\theta \in E \text{ and } b'_1(\theta, y^1) = 1) \mid b_1 = 1]$$

$$+ 0.5 \cdot \underset{y^1, y^0, b_1, \theta \sim M(D), \theta' \sim M(D')}{\mathrm{E}} [I(\theta \in E \text{ and } b'_1(\theta, y^0) = 0) \mid b_1 = 0]$$

$$\geq 0.5 \cdot \left( f^{-1}\left( 1 - \underset{y^1, y^0, b_1, \theta \sim M(D), \theta' \sim M(D')}{\mathrm{E}} [I(\theta' \in E \text{ and } b'_1(\theta', y^1) = 1) \mid b_1 = 1] \right) \right)$$

$$+ 0.5 \cdot \left( f^{-1}\left( 1 - \underset{y^1, y^0, b_1, \theta \sim M(D), \theta' \sim M(D')}{\mathrm{E}} [I(\theta' \in E \text{ and } b'_1(\theta', y^0) = 0) \mid b_1 = 0] \right) \right)$$

$$\geq f^{-1}(1 - \underset{y^1, y^0, b_1, \theta \sim M(D), \theta' \sim M(D')}{\mathrm{E}} [I(\theta' \in E \text{ and } b_1 \neq b'_1)]) \quad \text{(By convexity of } f.)$$

$$= f^{-1}(1 - q + p + \tau) \quad \text{(By the fact that } f^{-1} \text{ is decreasing.)}.$$

This implies $p \geq f^{-1}(1 - q + p)$ which in turn implies $f(p) + q - p + \tau \leq 1$. So we have $p \geq f''(q)$. □

We can now plug Lemma 10 in the proof of Theorem 3.2 in Mahloujifar et al. (2025b). For completeness, we state and prove all the steps from Mahloujifar et al. (2025b) although they are highly similar.

**Proposition 11.** *The functions $f'_\tau$ as defined in Lemma 10 is increasing and concave. The functions $f''_\tau$ as defined in Lemma 10 is increasing and convex.*

*Proof.* Please refer to the proof of Proposition A.2 in Mahloujifar et al. (2025a). □

Now we prove the following theorem which imposes a recursive relation on the number of correct guesses.

**Theorem 12.** *Let $M : (\mathcal{X} \times \mathcal{Y})^* \to \mathcal{Z}$ be a mechanism, $\mathcal{D}$ a distribution on $\mathcal{X} \times \mathcal{Y}$ and $\mathrm{Sim}_{M, \mathcal{D}'}$ the imputation-based simulator (Definition 2) and $\forall x \in \mathcal{X}, \mathrm{TV}(\mathcal{D}' \mid x, \mathcal{D} \mid x) \leq \tau$. Let $A : \mathcal{Z} \to \{0, 1, \perp\}^m$ be an attacker which always makes at most $c'$ guesses, that is*

$$\forall z \in \mathcal{Z}, \Pr\left[ \left( \sum_{i=1}^m I(A(z)_i \neq \perp) \right) > c' \right] = 0,$$

*and let $\mathbf{b}$ and $\mathbf{b}'$ be the random vectors of the observational attribute inference game defined in Algorithm 2 when instantiated with attacker $A$. Define $p_i = \Pr\left[ \left( \sum_{j \in [m]} \mathbf{I}(b_j = b'_j) \right) = i \right]$. If $M$ is f-SIM-DP with respect to $\mathrm{Sim}_{M, \mathcal{D}'}$, for all subsets of indices $T \subseteq [c']$, we have*

$$\sum_{i \in T} \frac{i}{m} p_i \leq 1 - f(\tau + \sum_{i \in T} \frac{c' - i + 1}{m} p_{i-1}).$$

*Proof of Theorem 12.* Instead of working with an adversary with $c'$ guesses, we assume we have an adversary that makes a guess on all $m$ inputs, however, it also submits a vector $\mathbf{q} \in \{0, 1\}^m$, with exactly $c'$ 1s and $m - c'$ 0s. So the output of this adversary is a vector $\mathbf{b}' \in \{0, 1\}^m$ and a vector $\mathbf{q} \in \{0, 1\}^m$. Then, only correct guesses that are in locations that $\mathbf{q}$ is non-zero are counted. That is, if we define a random variable $\mathbf{t} = (\mathbf{t}_1, \ldots, \mathbf{t}_m)$ as $\mathbf{t}_i = \mathbf{I}(\mathbf{b}_i = \mathbf{b}'_i) \cdot \mathbf{q}_i$ then we have

$$\sum_{j \in T} p_j = \sum_{j \in T} \Pr\left[\sum_{i=1}^{m} \mathbf{t}_i = j\right]$$

$$= \sum_{j \in T} \Pr\left[\sum_{i=2}^{m} \mathbf{t}_i = j \text{ and } \mathbf{t}_1 = 0\right] + \sum_{j \in T} \Pr[\sum_{i=2}^{m} \mathbf{t}_i = j - 1 \text{ and } \mathbf{t}_1 = 1]$$

$$= \Pr[\sum_{i=2}^{m} \mathbf{t}_i \in T \text{ and } \mathbf{t}_1 = 0] + \Pr[1 + \sum_{i=2}^{m} \mathbf{t}_i \in T \text{ and } \mathbf{t}_1 = 1]$$

$$= \Pr[\sum_{i=2}^{m} \mathbf{t}_i \in T \text{ and } \mathbf{t}_1 = 0] + \Pr[1 + \sum_{i=2}^{m} \mathbf{t}_i \in T \text{ and } \mathbf{b_1} = \mathbf{b'_1} \text{ and } \mathbf{q}_1 = 1]$$

Now we only use the inequality from Lemma 10 for the second quantity above. Using the inequality for both probabilities is not ideal because they cannot be tight at the same time. So we have,

$$\sum_{j \in T} p_j \leq \Pr[\sum_{i=2}^{m} \in T \text{ and } \mathbf{t}_1] + f'_\tau(\Pr[1 + \sum_{i=2}^{m} \mathbf{t}_i \in T \text{ and } \mathbf{q}_1 = 1]).$$

Now we use the fact that this inequality is invariant to the order of indices. So we can permute $\mathbf{t_i}$'s and the inequality still holds. We have,

$$\sum_{j \in T} p_j \leq \operatorname*{E}_{\pi \sim \Pi[m]}[\Pr[\sum_{i=2}^{m} \mathbf{t}_{\pi(i)} \in T \text{ and } \mathbf{t}_{\pi(1)} = 0]] + \operatorname*{E}_{\pi \sim \Pi[m]}[f'_\tau(\Pr[1 + \sum_{i=2}^{m} \mathbf{t}_{\pi(i)} \in T \text{ and } \mathbf{q}_{\pi(1)} = 1])]$$

$$\leq \operatorname*{E}_{\pi \sim \Pi[m]}[\Pr[\sum_{i=2}^{m} \mathbf{t}_{\pi(i)} \in T \text{ and } \mathbf{t}_{\pi(1)} = 0]] + f'_\tau(\operatorname*{E}_{\pi \sim \Pi[m]}[\Pr[1 + \sum_{i=2}^{m} \mathbf{t}_{\pi(i)} \in T \text{ and } \mathbf{q}_{\pi(1)} = 1]]).$$

Now we perform a double counting argument. Note that when we permute the order $\sum_{i=2}^{m} \mathbf{t}_{\pi(i)} = j$ and $\mathbf{t}_{\pi(1)} = 0$ counts each instance $t_1, \ldots, t_m$ with exactly $j$ non-zero locations, for exactly $(m - j) \times (m - 1)!$ times. Therefore, we have

$$\operatorname*{E}_{\pi \sim \Pi[m]}[\Pr[\sum_{i=2}^{m} \mathbf{t}_{\pi(i)} \in T \text{ and } \mathbf{t}_{\pi(1)} = 0]] = \sum_{j \in T} \frac{m - j}{m} p_j.$$

With a similar argument we have,

$$\operatorname*{E}_{\pi \sim \Pi[m]}[\Pr[1 + \sum_{i=2}^{m} \mathbf{t}_{\pi(i)} \in T \text{ and } \mathbf{q}_{\pi(1)} = 1]] = \sum_{j \in T} \frac{c' - j + 1}{m} p_{j-1} + \frac{j}{m} p_j.$$

Then, we have

$$\sum_{j \in T} p_j \leq \sum_{j \in T} \frac{m - j}{m} p_j + f'_\tau(\sum_{j \in T} \frac{j}{m} p_j + \frac{c' - j + 1}{m} p_{j-1})$$

$$= \sum_{j \in T} \frac{m - j}{m} p_j + f'_\tau(\sum_{j \in T} \frac{j}{m} p_j + \frac{c' - j + 1}{m} p_{j-1}).$$

And this implies

$$\sum_{j \in T} \frac{j}{m} p_j \leq f'_\tau(\sum_{j \in T} \frac{j}{m} p_j + \frac{c' - j + 1}{m} p_{j-1}).$$

And this, by definition of $f'_\tau$ implies

$$\sum_{j \in T} \frac{j}{m} p_j \leq 1 - f(\sum_{j \in T} \frac{c' - j + 1}{m} p_{j-1} + \tau).$$

$\square$

We now state and prove a lemma implied by Theorem 12.

**Lemma 13.** *For all $c \leq c' \in [m]$ let us define*

$$\alpha_c = \sum_{i=c}^{c'} \frac{i}{m} p_i \quad and \quad \beta_c = \sum_{i=c}^{c'} \frac{c'-i}{m} p_i$$

*We also define a family of functions $r = \{r_{i,j} : [0,1] \times [0,1] \to [0,1]\}_{i \leq j \in [m]}$ and $h = \{h_{i,j} : [0,1] \to [0,1]\}$ that are defined recursively as follows. Assume $\bar{f}(s) = 1 - f(s+\tau)$ for $s \in [0, 1-\tau]$ and also $\bar{f}^{-1}(r) = f^{-1}(1-r) - \tau$. Now define $\forall i \in [m] : r_{i,i}(\alpha, \beta) = \alpha$ and $h_{i,i}(\alpha, \beta) = \beta$ and for all $i < j$ we have*

$$h_{i,j}(\alpha, \beta) = \bar{f}^{-1}\Big(r_{i+1,j}(\alpha, \beta)\Big)$$

$$r_{i,j}(\alpha, \beta) = r_{i+1,j}(\alpha, \beta) + \frac{i}{c'-i}(h_{i,j}(\alpha, \beta) - h_{i+1,j}(\alpha, \beta))$$

*Then for all $i \leq j$ we have*

$$\alpha_i \geq r_{i,j}(\alpha_j, \beta_j) \quad and \quad \beta_i \geq h_{i,j}(\alpha_j, \beta_j)$$

*Moreover, for $i < j$, $r_{i,j}$ and $h_{i,j}$ are increasing with respect to their first argument and decreasing with respect to their second argument.*

*Proof of Lemma 13.* We prove this by induction on $j - i$. For $j - i = 0$, the statement is trivially correct. We have

$$h_{i,j}(\alpha_j, \beta_j) = \bar{f}^{-1}(r_{i+1,j}(\alpha_j, \beta_j)).$$

By the induction hypothesis, we have $r_{i+1,j}(\alpha_j, \beta_j) \leq \alpha_{i+1}$. Therefore we have

$$h_{i,j}(\alpha_j, \beta_j) \leq \bar{f}^{-1}(\alpha_{i+1}). \tag{3}$$

Now by invoking Theorem 12, we have

$$\alpha_{i+1} \leq \bar{f}(\beta_i).$$

Now since $\bar{f}$ is increasing, this implies

$$\bar{f}^{-1}(\alpha_{i+1}) \leq \beta_i \tag{4}$$

Now putting, inequalities 3 and 4 together we have $h_{i,j}(\alpha_j, \beta_j) \leq \beta_i$. This proves the first part of the induction hypothesis for the function $h$. The function $h_{i,j}$ is increasing in its first component and decreasing in the second component by invoking the induction hypothesis and the fact that $\bar{f}^{-1}$ is increasing. Consider the function $r_{i,j}$. There is an alternative form for $r_{i,j}$ by opening up the recursive relation. Let $\gamma_z = \frac{z}{c'-z} - \frac{z-1}{c'-z+1}$. We have ,

$$r_{i,j}(\alpha, \beta) = r_{j,j}(\alpha, \beta) + \frac{i}{c'-i}h_{i,j}(\alpha, \beta) - \frac{j-1}{c'-j+1}h_{j,j}(\alpha, \beta) + \sum_{z=i+1}^{j-1} \gamma_z h_{z,j}(\alpha, \beta)$$

$$= r_{j,j}(\alpha, \beta) + \frac{i}{c'-i}h_{i,j}(\alpha, \beta) - \frac{j}{c'-j}h_{j,j}(\alpha, \beta) + \sum_{z=i+1}^{j} \gamma_z h_{z,j}(\alpha, \beta)$$

$$= \alpha - \frac{j}{c'-j}\beta + \frac{i}{c'-i}h_{i,j}(\alpha, \beta) + \sum_{z=i+1}^{j} \gamma_z h_{z,j}(\alpha, \beta). \tag{5}$$

Now observe that for all $i$ we have

$$\alpha_i = \frac{i}{c'-i}\beta_i + \sum_{z=i+1}^{m} \gamma_z \beta_z. \tag{6}$$

Therefore for all $i < j$ we have

$$\alpha_i - \alpha_j = \frac{i}{c' - i}\beta_i - \frac{j}{c' - j}\beta_j + \sum_{z=i+1}^{j} \gamma_z \beta_z$$

Now, using the induction hypothesis for $h$ we have,

$$\alpha_i \geq \alpha_j + \frac{i}{c' - i}h_{i,j}(\alpha_j, \beta_j) - \frac{j}{c' - j}\beta_j + \sum_{z=i+1}^{j} \gamma_z h_{z,j}(\alpha_j, \beta_j). \tag{7}$$

Now verify that the right hand side of Equation 7 is equal to $r_{i,j}(\alpha_j, \beta_j)$ by the formulation of Equation 5

Also, using the induction hypothesis, we can observe that the right hand side of 5 is increasing in $\alpha_j$ and decreasing in $\beta_j$ because all terms there are increasing in $\alpha_j$ and decreasing in $\beta_j$. $\qquad\square$

Now assume that the probability of correctly guessing $c$ examples out of $c'$ guesses is equal to $\tau'$. Namely, $\alpha_c + \beta_c = \frac{c'}{m}\tau'$. Note that

$$\frac{\alpha_c}{\beta_c} \geq \frac{c}{c' - c} \tag{8}$$

therefore, we have

$$\alpha_c \geq \frac{c}{m}\tau' \tag{9}$$

and $\beta_c \leq \frac{c'-c}{m}\tau'$. Therefore, using Lemma 10 we have $\alpha_0 \geq r_{0,c}(\frac{c}{m}\tau', \frac{c'-c}{m}\tau')$ and $\beta_0 \geq h_{0,c}(\frac{c}{m}\tau', \frac{c'-c}{m}\tau')$.

Now we prove a lemma about the function $s_{i,j}(\tau) = h_{i,j}(\frac{c}{m}\tau, \frac{c'-c}{m}\tau) + r_{i,j}(\frac{c}{m}\tau, \frac{c'-c}{m}\tau)$.

**Lemma 14.** *The function* $s_{i,j}(\tau) = h_{i,j}(\frac{c}{m}\tau, \frac{c'-c}{m}\tau) + r_{i,j}(\frac{c}{m}\tau, \frac{c'-c}{m}\tau)$ *is increasing in* $\tau$ *for* $i < j \leq c$.

*Proof.* To prove this, we show that for all $i < j \leq c$ both $r_{i,j}(\frac{c}{m}\tau, \frac{c'-c}{m}\tau)$ and $h_{i,j}(\frac{c}{m}\tau, \frac{c'-c}{m}\tau)$ are increasing in $\tau$. We prove this by induction on $j - i$. For $j - i = 1$, we have

$$h_{i,i+1}(\frac{c}{m}\tau, \frac{c'-c}{m}\tau) = (k-1)\bar{f}^{-1}(\frac{c}{m}\tau).$$

We know that $\bar{f}^{-1}$ is increasing, therefore $h_{i,i+1}(\frac{c}{m}\tau, \frac{c'-c}{m}\tau)$ is increasing in $\tau$ as well. For $r_{i,i+1}$ we have

$$r_{i,i+1}(\frac{c}{m}\tau, \frac{c'-c}{m}\tau) = \frac{c}{m}\tau + \frac{i}{c'-i}(h_{i,i+1}(\frac{c}{m}\tau, \frac{c'-c}{m}\tau) - \frac{c'-c}{m}\tau).$$

So we have

$$r_{i,i+1}(\frac{c}{m}\tau, \frac{c'-c}{m}\tau) = \frac{c(c'-i) - i(c'-c)}{m(c'-i)}\tau + \frac{i}{c'-i}h_{i,i+1}(\frac{c}{m}\tau, \frac{c'-c}{m}\tau)$$

$$= \frac{(c-i)c'}{m(c'-i)}\tau + \frac{i}{c'-i}h_{i,i+1}(\frac{c}{m}\tau, \frac{c'-c}{m}\tau).$$

We already proved that $h_{i,i+1}(\frac{c}{m}\tau, \frac{c'-c}{m}\tau)$ is increasing in $\tau$. We also have $\frac{(c-i)c'}{m(c'-i)} > 0$, since $i < c$. Therefore

$$r_{i,i+1}(\frac{c}{m}\tau, \frac{c'-c}{m}\tau)$$

is increasing in $\tau$. So the base of the induction is proved. Now we focus on $j - i > 1$. For $h_{i,j}$ we have

$$h_{i,j}(\frac{c}{m}\tau, \frac{c'-c}{m}\tau) = (k-1)\bar{f}^{-1}(r_{i+1,j}(\frac{c}{m}\tau, \frac{c'-c}{m}\tau)).$$

By the induction hypothesis, we know that $r_{i+1,j}(\frac{c}{m}\tau, \frac{c'-c}{m}\tau)$ is increasing in $\tau$, and we know that $\bar{f}^{-1}$ is increasing, therefore, $h_{i,j}(\frac{c}{m}\tau, \frac{c'-c}{m}\tau)$ is increasing in $\tau$.

For $r_{i,j}$, note that we rewrite it as follows

$$r_{i,j}(\alpha, \beta) = \alpha - \frac{j}{c'-j}\beta + \sum_{z=i}^{j-1} \lambda_z \cdot h_{z,j}(\alpha, \beta)$$

where $\lambda_z = (\frac{z+1}{c'-z-1} - \frac{z}{c'-z}) \geq 0$. Therefore, we have

$$r_{i,j}(\frac{c}{m}\tau, \frac{c'-c}{m}\tau) = \tau(\frac{c}{m} - \frac{(c'-c)j}{m(c'-j)}) + \sum_{z=i}^{j-1} \lambda_z \cdot h_{z,j}(\frac{c}{m}\tau, \frac{c'-c}{m}\tau)$$

$$= \tau \frac{c'(c-j)}{m(c'-j)} + \sum_{z=i}^{j-1} \lambda_z \cdot h_{z,j}(\frac{c}{m}\tau, \frac{c'-c}{m}\tau).$$

Now we can verify that all terms in this equation are increasing in $\tau$, following the induction hypothesis and the fact that $\lambda_z > 0$ and also $j \leq c$. $\qquad\square$

Now using this Lemma, we finish the proof. Note that we have $\alpha_0 + \beta_0 = \frac{c'}{m}$.

Assuming that $\tau' \geq \tau$, then we have

$$\frac{c'}{m} = \alpha_0 + \beta_0 \geq s_{0,c}(\tau') \geq s_{0,c}(\tau).$$

The last step of the algorithm checks whether $s_{0,c} \geq \frac{c'}{m}$. If so, then $\tau' \leq \tau$, because $s_{0,c}$ is increasing in $\tau$. This means that the probability of having more than $c$ guesses cannot be more than $\tau$. $\qquad\square$

## C  AUDITING RANDOMIZED RESPONSE

We empirically demonstrate the tightness of our Label DP auditing algorithm for Randomized Response on synthetic data sampled from a mixture of Gaussian distributions.

We experiment with binary and multi-class labels. The labels $y$ are sampled from Multinoulli($k, (\frac{1}{k}, \ldots, \frac{1}{k})$), a distribution with $k$ classes and equal probabilities for each class. Given a label $y$, the features $x$ are sampled according to:

$$x \mid y \sim \mathcal{N}(e_y, I_d), \tag{10}$$

where $I_d$ is the $d$-dimensional identity matrix, and $e_y$ is a $d$-dimensional index vector that is 1 at index $y$.

The output of the Randomized Response mechanism are the noisy labels. Recall that $y_i^0$ is the training set label and $y_i^1$ is the reconstructed label. We generate $y_i^1$ in two ways:

- Ground truth: By using the posterior $y \mid x$ on the label, for $y$ generated as in (10).
- Proxy: By using the predictions of a Logistic Regression model trained on fresh data from the distribution (with default hyper-parameters from scikit-learn).

This way, we compare the effect of using a proxy distribution instead of the ground truth for generating the labels $y_i^1$. In Figures 2 and 3, we see that the two methods for generating $y^1$ yield similar results, as Logistic Regression can approximate a mixture of Gaussians quite well.

We use $n = 10^6$, $d = 5$, and the number of classes $k \in \{2, 5, 10\}$. We run our attack on the output of Randomized Response and with a fraction of non-abstaining guesses in $\{0.1\%, 1\%\}$. We use Theorem 5 with $\tau = 0$ to obtain the empirical epsilon achieved at 95% confidence. For each dataset and Randomized Response output, we repeat the game $10^2$ times (with a fresh vector of reconstructed labels) and compute the average and standard deviation of the empirical epsilon.

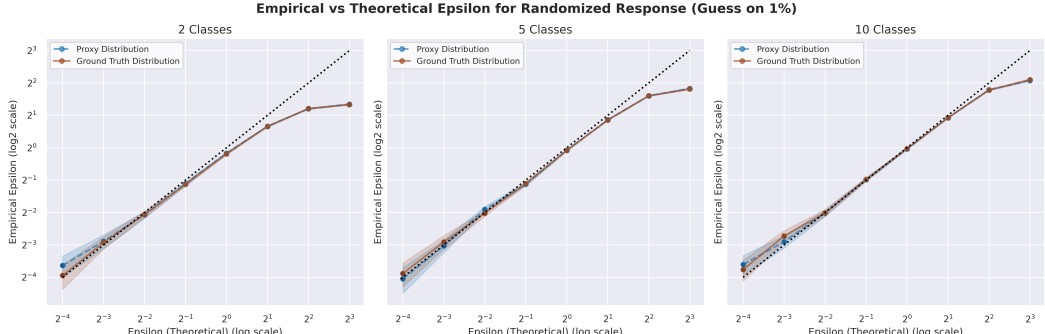

Figure 3: Auditing randomized response when the adversary guesses on 1% of samples. The counterfactual labels are generated either from the ground-truth distribution or a proxy distribution yielded by the Logistic Regression model.

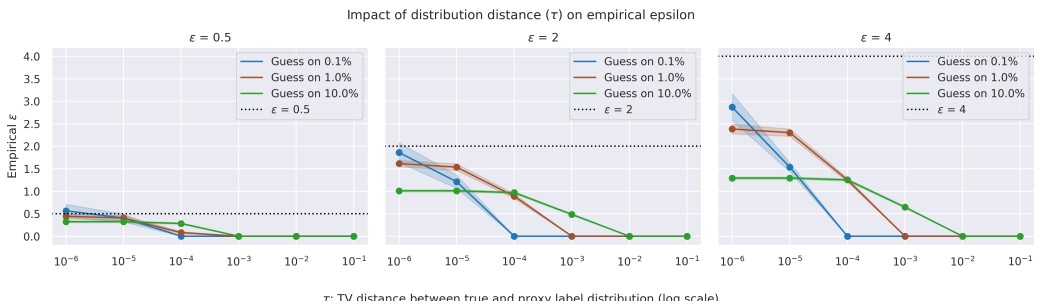

Figure 4: Auditing randomized response when synthetic labels are produced from a $\tau$-shifted distribution from the ground truth label distribution.

Fig. 2 showed that with (fewer) $0.1\%$ adversarial guesses we overestimate the privacy loss at small theoretical epsilon, but the lower bound is tight with (more) $1\%$ guesses in the low-epsilon regime (Fig. 3). However, the lower bounds are not as tight in the medium epsilon regime $[1, 4]$. These experiments show that choosing the number of adversarial guesses should take into consideration the privacy regime we are targeting.

**Impact of distribution shift $\tau$.** To demonstrate the impact of the distribution shift $\tau$ on the measured empirical epsilon, we modify our synthetic experiments on randomized response so that proxy labels are generated for a *shifted* ground truth label posterior. We focus on the binary case, where training labels $y^0 \in \{0, 1\}$. We produce proxy labels as

$$y^1 \sim \text{Bernoulli}(s_x^\tau), \text{where } s_x = \min(\Pr[y = 1 \mid x] + \tau, 1).$$

Note that $\text{TV}(\text{Bernoulli}(s_x^\tau), y \mid x) \leq \tau$.

We vary the theoretical $\epsilon \in \{0.5, 2, 4\}$ and $\tau \in \{10^{-6}, 10^{-5}, \ldots, 10^{-1}\}$, and measure empirical $\epsilon$ according to Theorem 5 at different guess fractions for the adversary. As for other experiments, $\epsilon$ is computed at $95\%$ confidence and standard deviation is computed over 100 game repetitions.

In Fig. 4 we see that as expected, for larger values of $\tau$ the empirical epsilon measured decreases, i.e., decreasing the quality of the simulator decreases the privacy leakage that can be measured. At $\tau \geq 10^{-2}$ the empirical epsilon is near 0. This is a limitation of the bound of Theorem 5, which has an additive dependence on $\tau$. Allowing the adversary to make more guesses at larger $\tau$ can ameliorate gap between the measured empirical $\epsilon$ and theoretical $\epsilon$.

Table 4: CIFAR-10. Model test accuracy of Label DP models under different $\epsilon$.

| Label DP Algorithm | CIFAR-10 | | |
|---|---|---|---|
| | $\epsilon = \infty$ | $\epsilon = 10.0$ | $\epsilon = 1.0$ |
| LP-1ST | 91.3 | 91.07 | 60.4 |
| LP-1ST (out-of-domain prior) | 92.1 | 91.5 | 87.9 |
| PATE-FM | 92.5 | 92.3 | 91.3 |
| ALIBI | 90.1 | 87.1 | 66.9 |

Table 5: Criteo. Log-loss of Label DP algorithms on the test set under different $\epsilon$.

| Label DP Algorithm | $\epsilon = \infty$ | $\epsilon = 8$ | $\epsilon = 4$ | $\epsilon = 2$ | $\epsilon = 1$ | $\epsilon = 0.1$ |
|---|---|---|---|---|---|---|
| LP-1ST | 0.130 | 0.130 | 0.136 | 0.206 | 0.362 | 0.653 |
| LP-1ST (domain prior) | 0.130 | 0.130 | 0.136 | - | - | - |
| LP-1ST (noise correction) | 0.130 | 0.130 | 0.131 | 0.156 | 0.171 | 0.645 |
| LP-2ST | 0.130 | 0.130 | 0.123 | 0.207 | 0.342 | 0.527 |
| PATE | 0.130 | 0.151 | 0.156 | 0.188 | 0.255 | 0.680 |

# D  ACCURACY OF MODELS TRAINED ON CRITEO AND CIFAR-10

Tables 4 and 5 show the performance of models trained on CIFAR-10 and Criteo, respectively, for different Label DP algorithms and privacy budgets $\epsilon$.

# E  ADDITIONAL EXPERIMENTS

**Fixed fraction of adversary guesses.**  In our CIFAR-10 and Criteo experiments, we sweep over the fraction of adversarial guesses $c' \in \{1\%, 2\%, \ldots, 100\%\}$ and report the highest empirical $\epsilon$. This can overestimate empirical $\epsilon$ as it does not account for multiple hypothesis testing (albeit, the hypothesis in this case are highly correlated). In Table 6, we show the results on CIFAR-10, where we fix the fraction of adversary guesses to $c' = 1\%$ of samples. Compared to Table 1 where we sweep $c' \in \{1, 2, \ldots, 100\}$, we observe slightly smaller $\epsilon$ values.

Table 6: CIFAR-10. Auditing Label DP algorithms under different $\epsilon$ with $\delta = 10^{-5}$ and a fixed fraction of $1\%$ adversarial guesses.

| Label DP Algorithm | $\epsilon = \infty$ | $\epsilon = 10.0$ | $\epsilon = 1.0$ |
|---|---|---|---|
| LP-1ST | $1.98 \pm 0.21$ | $1.99 \pm 0.32$ | $0.41 \pm 0.03$ |
| LP-1ST (out-of-domain prior) | $2.08 \pm 0.21$ | $1.78 \pm 0.23$ | $0.81 \pm 0.05$ |
| PATE-FM | $2.26 \pm 0.31$ | $2.16 \pm 0.22$ | $0.74 \pm 0.08$ |
| ALIBI | $2.43 \pm 0.31$ | $2.1 \pm 0.26$ | $0.65 \pm 0.05$ |

**Comparison to additional prior MIA methods.**  In Fig. 5, we report the performance of the Robust MIA (RMIA) approach from Zarifzadeh et al. (2024). This attack is designed to have high power when training very few additional models for the attack. We run the offline mode of RMIA with the $M'$ model as the single shadow (reference) model for the non-members data distribution. We observe that RMIA slightly outperforms both the observational LIA and the *difficulty calibration MIA* (Watson et al., 2022), particularly for large theoretical $\epsilon$ regimes.

**Additional dataset.**  We use the *Twitter Sentiment* dataset Awasthi (2023) with 3 million tweets. We remove the small number of neutral tweets from the dataset and keep the (balanced) positive and negative labeled tweets. We keep a random 10% portion of the dataset as test set and randomly split the remaining data into two. We train the reference $M'$ on the first half, and the target model on the other half. We vectorized the tweets using TF-IDF features and train standard Fully Connected

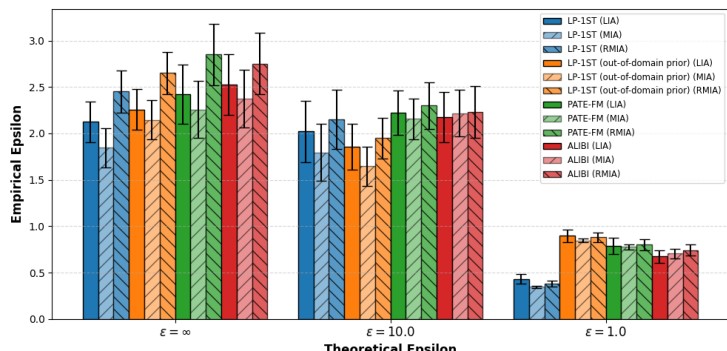

Figure 5: Comparison with calibration-based MIA Watson et al. (2022) and RMIA Zarifzadeh et al. (2024) for different Label DP Algorithms on CIFAR-10 dataset. The error bar represents the standard deviation across 100 different attack repetitions.

Table 7: Auditing a model trained on the Twitter sentiment analysis dataset (with $\epsilon = \infty$).

| Attack | Empirical $\epsilon$ ($\pm$ standard deviation) |
|---|---|
| Observational LIA (ours) | $2.34 \pm 0.25$ |
| Calibration-based MIA (Watson et al., 2022) | $0.72 \pm 0.15$ |
| RMIA (Zarifzadeh et al., 2024) | $2.52 \pm 0.32$ |

Neural Networks to achieve $80\%$ accuracy on the test set. We use $300K$ canaries for LIA from the training set and the same number of canaries from the test set for the MIA experiments. In Table 7 we report the auditing results on the non-private model (theoretical $\epsilon = \infty$).

