# OpenReview forum: "Observational Auditing of Privacy"
_ICLR.cc/2026/Conference — ICLR 2026 Conference Desk Rejected Submission_

### Official Review · Reviewer_5SJ9 · 2025-10-25

**Soundness:** 3
**Presentation:** 2
**Contribution:** 2
**Rating:** 2
**Confidence:** 5

**Summary:**

This paper introduces an observational privacy auditing framework for evaluating differential privacy (DP) guarantees without modifying or re-running the training pipeline. Unlike interventional audits that inject canaries or retrain models on modified datasets, the proposed approach leverages natural variation in the data and a proxy label-generating distribution to construct a post-training attribute/label inference game. The framework is formalized through simulation-based DP for protected attributes, an imputation-based simulator, and an observational attribute inference game (Algorithm 2). Theoretical results provide auditing guarantees both in the ideal case (no distribution shift) and under bounded distributionla shift between the true and proxy distributions. Experiments on CIFAR-10 and Criteo illustrate that the method is competitive with interventional baselines while requiring no training-time intervention. Overall, the paper aims to lower the engineering barrier of auditing Label-DP mechanisms in practical settings.

**Strengths:**

- **Practical relevance and usability:** The approach avoids training-time interventions (no canaries, no retraining), which significantly lowers engineering overhead and makes the audit feasible for third-party or post-hoc evaluation settings.
- **Clear formalization of observational auditing:** The use of simulation-based DP for protected attributes, together with the imputation-based simulator and the observational game, provides a principled lens for analyzing privacy leakage without data modification.
- **Addresses real-world setting of distribution shift:** The theoretical treatment of bounded distribution shift is both interesting and important, as it reflects realistic deployment scenarios where training and proxy distributions may drift.
- **Empirical evaluation is fairly comprehensive:** The paper covers two qualitatively different domains (vision and large-scale tabular data) and several Label-DP mechanisms, and includes synthetic randomized response experiments to illustrate tightness and limitations.
- **Potential impact for practice:** If validated further, the technique could make privacy auditing more accessible to practitioners who cannot modify training workflows or re-run large models.

**Weaknesses:**

- **Scope of presentation regarding Label-DP vs. general DP:**  The paper focuses on Label-DP, a more narrow neighboring relation. While the framework may extend more broadly, the current exposition remains Label-DP centric. Providing more explicit guidance on how it generalizes to standard record-level DP could broaden its relevance.

- **Choice of presenting both $(\varepsilon,\delta)$-DP and $f$-DP results:**   Both formalisms are presented, but the benefits of including both are not fully explained. Since $f$-DP can subsume $(\varepsilon,\delta)$: it is well known that by taking the convex conjugate of an $f$-DP, you can recover the $\epsilon, \delta(\epsilon)$ curve, see Dong et al. (2022) Proposition 2.12. Furthermore, the authors provide proof only for the $(\epsilon, 0)$ case, claiming that the $(\epsilon, \delta)$ proof follows from the original proof. I would personally stick with the f-DP and present it in detail, as it is essentially duplicating information without additional value for the reader.

- **Unclear impact of $M'$** : The impact of $M'$ is largely underexplored experimentally. What does it mean to have a "good" $M'$, what's the impact of not having a good enough one and how it reflects into the lowerbound is largely undiscussed.

- **Assumptions required for observational auditing are not clearly highlighted:**   Beyond access to a proxy distribution $D'$, the implicit assumptions (e.g., on the adversary’s knowledge, calibration/quality of $M'$, independence assumptions) are not clearly stated. A concise assumptions summary would make the framework more transparent.

- **Positioning relative to prior “security game” formulations:**  The paper claims to reframe auditing as a security game, but prior works  also use game-based formulations. Clarifying what is novel in this framing would avoid confusion.

- **Generality claim around neighboring relations (Lines 143–147):**  The claim that the simulator game generalizes add/remove, leave-one-out, or zero-out relations could be seen as overstated. The simulator still implicitly encodes a neighboring relation, this could be clarified.

- **Heavy reliance on background knowledge:**   The paper depends on familiarity with ideas from Steinke et al. (2024) and Mahloujifar (2025). Brief intuitive overviews would help make the paper more self-contained, for example, how refusing to decide on the membership actually aids auditing.

- **Explaining observed gaps between audit and theoretical upper bounds:**  The paper does not sufficiently discuss why the empirical lower bounds are sometimes noticeably below theoretical ones. Additional interpretation would benefit readers.

- **Assumptions for $f$-DP auditing prior:**  For f-DP auditing, you mention you pick as a prior that you expect to see a Gaussian DP tradeoff curve. How is this valid/justified? For auditing a single subsampled mechanism or a pure-DP mechanism, this is clearly incorrect. Furthermore, you do not clearly highlight the need for this before the appendix (lines 735-740). This seems an essential note for your auditing technique. Furthermore, if I want to audit the composition of a Gaussian and an RR mechanism, there is no clear/known structure you could know beforehand (assuming that the adversary only observes the output of some composition). Can the authors clarify this?

- **Unclear behaviour under distributional shifts:** From my point of view, the central contribution of this paper is actually handling the distribution shifts. Still, it is a bit unclear how to integrate this idea into practice, and the paper does not fully explore it. Given some practical scenario, it is unclear how to upperbound the distributional shift, as we don't have access to $D | x$. The paper does not provide practical estimators, proxies, or diagnostics to quantify the change between the real data and the proxy distribution. Also,  it would be helpful to show audit quality as a function of artificially increasing divergence (e.g., perturbing D′ in a controlled way).

Minor points
-  Some terms (e.g., “PPML”) appear without definition.
- Showing $(\varepsilon,\delta(\varepsilon))$ curves (or multiple $\delta$ values such as $10^{-1},10^{-2},10^{-3} \cdots)$ would provide a more complete picture, as suggested by Gómez et al. (2025).

- Algorithm 1 is quite abstract and introduces notation that is not really reused afterwards. It may be better suited for the appendix or supported with a running example to improve readability.

- Some Label-DP mechanisms are non-standard. A short summary or table of the mechanisms and their structure in the Appendix would aid reader understanding, especially for the experimental section.

- Clarity of paragraph around lines 085–090:  The messaging here is hard to follow and could benefit from restructuring for clarity.


Gomez et al., 2025: https://arxiv.org/abs/2503.10945

Dong et al., 2022: https://arxiv.org/abs/1905.02383

Nasr et al. 2021: https://arxiv.org/abs/2101.04535

**Questions:**

- What concrete steps are required to generalize this framework beyond Label-DP to standard record-level DP?  A short discussion or example would help clarify applicability beyond the label setting.

- Could the authors clarify the specific novelty of their security-game formulation compared to earlier works such as Nasr et al. 2021?

- Can the authors explicitly list all assumptions required for their observational auditing guarantees (beyond access to a proxy distribution $D'$)?

- How should practitioners choose or evaluate the proxy model $M'$?   For example, how good does $M'$ need to be for the audit to provide a meaningful bound, and how can one detect when a proxy is too weak?

- The gap between the audit and the theoretical upper bound is sometimes significant in the experiment, but the authors provide minimal context for why that is happening. Is it because of the audit, or is the upper bound loose? Can other (that are not necessarily one-run) auditing techniques perform better? If yes, how well? All these questions should be clarified.

- Is it possible to avoid assuming a Gaussian prior for $f$-DP auditing, or to provide alternatives for mechanisms where Gaussianity is not a natural fit? (see the Weaknesses section)

- How does your guarantee behave under different numbers of abstentions from a decision by the adversary?

---

> ### Author Response · Authors · 2025-11-21
>
> Thank you for your detailed comments and questions and the valuable input. We address the questions first.
>
> **Generalizing beyond label inference attacks**
> >What concrete steps are required to generalize this framework beyond Label-DP to standard record-level DP? A short discussion or example would help clarify applicability beyond the label setting.
>
> The only missing piece is being able to generate entire synthetic records (for observational MIA) or generate synthetic features conditioned on the public features (for observational attribute inference attacks). The approximation should be “good enough” in the sense that an adversary with realistic computational resources cannot distinguish between the synthetic and true samples a priori, without seeing the target model’s predictions. Generative models such as GANs or diffusion models are one such method to generate either entire samples or samples conditioned on other features. However, as opposed to generating labels, these methods require different architectures than the target model and would be more expensive to maintain in a production environment. This presents a tradeoff between ability to maintain training of additional models versus ability to modify the training pipeline. We agree that this merits further discussion in our paper and will include such discussion.
>
>
> **Novelty of the security game**
> >Could the authors clarify the specific novelty of their security-game formulation compared to earlier works such as Nasr et al. 2021?
>
> We formulate our security games to make explicit the fact that the adversary does not get to manipulate the training dataset. For instance, Nasr et al. 2021 consider scenarios where the adversary chooses a random (vs maliciously crafted) input. Even though an (equivalent) game where the canary is drawn from the training distribution \emph{can} be run observationally, it is not obvious from their presentation.
>
>
> **Assumptions required for observational auditing guarantees**
> >Can the authors explicitly list all assumptions required for their observational auditing guarantees (beyond access to a proxy distribution
> Our intention is to capture all assumptions through Algorithm 2 and include them in the main Theorems 5 and 6 through references to Algorithm 2. However, we can also state these assumptions explicitly:
> - The challenger has a dataset of points sampled iid from the distribution $\mathcal{D}$.
> - The challenger has access to a distribution $\mathcal{D}’$ supported on $\mathcal{X} \times \mathcal{Y}$ from which it can sample $y | x$.
> - The attacker has oracle access to the outputs of the mechanism and full access to $\mathcal{D}$ and $\mathcal{D}’$.
> - The total variation distance between $\mathcal{D}$ and $\mathcal{D’}$ is small compared to epsilon. More accurately, $\epsilon_\tau$ computed from $\beta$ according to Theorem 6 is approximately $\epsilon_0+2\tau$ for $\tau\ll 1$, where $\epsilon_0$ is derived from ``perfectly simulated’’ $\beta$.
>
> **Choosing the proxy model**
> >How should practitioners choose or evaluate the proxy model $M'$? And how should they account for the distribution shift?
> - In our discussion in lines 254-262 we provide one such guidance to practitioners: in a recurring (online) training setting, $M’$ can be a prior checkpoint to the target $M$, which eliminates the need for training additional models. This is akin to our Criteo experiments.
> - An interesting implication of Theorem 6 is that $M’$ should be the best model one can train to approximate the label distribution, which is compatible with the goal of the model owner. This supports our recommendation that $M’$ and $M$ should have the same architecture.
> - To elaborate, In lines 279-282 we discuss that Theorem 6 can also be stated in terms of the adversary’s ability to distinguish $M’(x)$ from $\mathcal{D}|x$ with realistic computational resources. A realistic adversary does not have exact knowledge of $y|x$ for a complex enough distribution $\mathcal{D}$, thus $M’$ should be the best model an adversary can train to estimate label probabilities.
> If the model owner is indeed training the most accurate model possible, then the above reframing of Theorem 6 allows practitioners to not have to account for the distribution shift. Beyond the label-DP setting, the model owner can estimate the attacker’s ability to distinguish between synthetic and true samples by fine-tuning a model that makes a binary real/synthetic prediction.

---

> > ### Author Response · Authors · 2025-11-21
> >
> > **Gap between audit and theoretical upper bounds**
> > >The gap between the audit and the theoretical upper bound is sometimes significant in the experiment, but the authors provide minimal context for why that is happening.
> >
> > The gaps are due to several factors: (1) the theoretical upper bound is not tight for the label-DP algorithms we attack, as a tight analysis is challenging, (2) the attack is black-box, e.g., white box MIA attacks achieve a 0.5 ratio between theoretical and empirical epsilon for CIFAR-10 compared to our ratios of ~0.3-0.5  (Figure 2 in https://arxiv.org/pdf/2410.22235). (3) Using multiple shadow runs would improve attack performance. Yet another reason is that the f-DP auditing method of Mahlojifar et al’25 is not tight for all mechanisms, with gaps that increase with the theoretical epsilon.
> >
> > **Priors for f-DP auditing**
> > >Is it possible to avoid assuming a Gaussian prior for f-DP auditing, or to provide alternatives for mechanisms where Gaussianity is not a natural fit? (see the Weaknesses section)
> >
> > We can always audit the privacy of a mechanism against any f-DP curve. It becomes more subtle when we want to calculate empirical epsilon. For this, we need to take a family of f-DP curves F and report the “best” curve that passes our audit. “Best” needs to be properly defined and we define it based on the value of epsilon at a given delta for that curve. Then when finding this curve, we need a way to represent it and we again use the value of epsilon at a given delta to report the curve. The theoretical interpretation of this empirical epsilon is as follows: if the privacy of the mechanism satisfies $f$-DP for any curve in the family of curves we have, then the value of epsilon at a given delta for that curve should be greater than empirical epsilon, with high probability. This point is also discussed the paragraph “How to choose the family of trade-off functions?” in https://arxiv.org/pdf/2410.22235.
> >
> > We also note that training algorithms that use batch-SGD with many steps behave like a gaussian mechanism. So it’s a good heuristic to assume gaussian mechanism as the family of trade-off functions for scenarios where we do not know the real trade-off function.
> >
> > Finally, f-DP auditing is only a generalization of DP auditing. We can always consider the family of trade-off functions associated with $(\\epsilon,\\delta)$-DP.
> >
> > **Number of abstentions**
> > >How does your guarantee behave under different numbers of abstentions from a decision by the adversary?
> >
> > Empirically, we explore this in Figure 2 and Figure 3, where the behavior depends on the epsilon regime. Very few guesses (many abstentions) can lead to wide confidence intervals in our case an overestimation of the theoretical epsilon in the low-epsilon regime, while underestimating in the medium epsilon regime [1, 4].  Conversely, with more guesses we underestimate in the medium epsilon regime but are tight in the low epsilon regime (lines 947-951). Finding the optimal value for the number of guesses is an interesting question. Currently both our and prior works (Steinke et al, Mahloujifar et al) iterate over many values and pick the maximum empirical epsilon achieved, without correcting for multiple hypothesis testing.
> >
> > We briefly go through other points raised by the reviewer
> >
> > **Choice of presenting both $(\\varepsilon,\\delta)$-DP and f-DP results**
> > We believe that the $(\\varepsilon,\\delta)$-DP results are easier to interpret in the case with distribution shift. There are no other advantages to using $(\\varepsilon,\\delta)$-DP over f-DP results, which give tighter auditing. We can move all discussion of  $(\\varepsilon,\\delta)$-DP to the appendix and focus on f-DP auditing in the main body.
> >
> > **Understanding behavior under distribution shift**
> > We will extend our experiments with synthetic data to scenarios where the proxy is artificially shifted from the true distribution.
> >
> > **Generality claim around neighboring relations (Lines 143–147)**
> > We do not claim simulation-based DP is our contribution; we merely state that it offers the equivalent (“generalization”) of zero-out DP for label/attribute-DP. We will rephrase the sentence for clarity.
> >
> > **Next steps**
> > To recap, our next steps are:
> >
> > - Consolidate and reiterate our guidance to practitioners for choosing M’ and accounting for distribution shift
> > - Extend our synthetic data experiments to illustrate the role of distribution shift
> > - Provide some examples for how the attack can extend beyond label inference
> > - Stick to f-DP auditing in the main body with (eps, delta)-results in the Appendix.
> > - Provide more explanation about the gap between theoretical and empirical results
> > - Provide a more detailed overview of the works of Steinke et al and Mahloujifar et al
> > - Provide more details about the label-DP attacks (in case this was overlooked we provide a description of each mechanism in Sec.6.1)

---

> > > ### Comment · Reviewer_5SJ9 · 2025-11-21
> > >
> > > If you could apply the above points and highlight the changes so I can do another pass, I am willing to increase my score to a weak accept. I think the paper's overall direction and ideas are interesting, but the main blocker is the exposition.

---

### Official Review · Reviewer_jL12 · 2025-10-25

**Soundness:** 2
**Presentation:** 1
**Contribution:** 2
**Rating:** 2
**Confidence:** 4

**Summary:**

Differential privacy (DP) is a privacy measure of a mechanism - a randomized function from a set of data points to some output. Its privacy is measured by the ability of an adversary to distinguish between two possible input datasets differing in a single element (referred to as neighbors), based on a single output of the mechanism. The better the privacy, the close is the success rate of the most sophisticated analyst to a random guess. This framing as an hypothesis testing naturally lends itself to an empirical lower bound scheme; select two neighboring datasets, repeatedly run the mechanism on one one of the two datasets selected at random, and guess the identity of the used dataset based on the mechanism's output. The more accurate the guesser, the lower the privacy. Recently, a more efficient method has emerged, where rather than repeating many time the experiment to determine a change in a single element, the mechanism is called once and the auditing attempts to identify the participation of many elements, but the general idea remains.

This work considers a combination of several modifications over the the undermentioned recent baseline. First, it considers the restrictive case where the training set cannot be altered, and instead the auditing is carried between elements independently sampled from the same distribution. Second, it generalizes to the case where the elements not used for training were not sampled from the same distribution but from a similar one. Finally, it generalizes the results from classical DP which is defined with respect to a change of a single element, to label-DP where part of the elements attributes are known, and the change applies only to a subset of features referred to as labels.

For this setting the authors provide theoretical guarantees which generalize previously known results, and report some empirical results.

**Strengths:**

All three generalization proposed by this work are natural useful goals, and this work sufficiently motivates them.
Unfortunately, I found the presentation confusing and the results either vacuous or suspicious (as detailed in the next section), so I find it hard to elaborate more on these strengths.

One clear contribution I've managed to extract from this work is the fact that in random elements-based auditing of label-DP, the resampling of the alternative labels can be done using a distribution close to the original one, for which the authors extended the lower bound guarantee by Steinke et al..

**Weaknesses:**

The main challenge I've experienced in understanding this work is the relation between the three generalization directions discussed above. In fact, I am not even sure the authors will agree with the way I represented their contribution as three separate generalizations, since they are represented in a way that gives the impression they are all interconnected, but I don't see that connection.

* From the problem setting perspective, I do not understand how does the supposedly new observational auditing method differ from a simple setting where $2 n$ points are sampled from $\mathcal{D}$, they are matched in pairs, from which one is added to the training set as random. The authors seem to consider the option of "excluding a subset of the data from training" as "severely restricting applications of these auditing methods", while their proposal is essentially identical. The only difference is that they represent it as a sampling of $n$ elements for the training set followed by sampling of $n$ additional elements, which is indistinguishable from sampling $2n$ elements and using a random subset of size $n$.
* On the theoretical level, the contribution is not clear to me. The Label DP aspect cannot affect the privacy guarantees, since the public features are akin to any other auxiliary information publicly known to the mechanism and auditor alike. The restriction to canaries sampled from the underlying distribution rather than adversarial constructed only weakens the auditing power, so I fail to understand what does what is the purpose of Theorem 5 if it is simply a special case of the known result by Steinke et al. (again, using the perspective where we first sampled $2n$ elements then selected a subset at random).
* On the experimental front I find the results surprising and somewhat suspicious. I will focus on the CIFAR10 experiment, which also appeared in previous works. Since both training and auditing examples are sampled according to the (nearly) same distribution, and the accuracy of the non-private model is $\approx 90\\%$, the public $x$ nearly determines the label $y$ so for $> 80\\%$ of data points the labels will be identical. Under such conditions, the auditing power of the random elements with known features $x$ must be significantly weaker than that of random elements where both $x$ and $y$ are resamples, but as figures 8 and 9 in Steinke et al. indicate, even changing the entire elements makes it very hard to achieve reasonable bounds, so I cannot understand how using only label DP with no adversarial canaries can reach such a (relatively) tight bound of $\epsilon=1$. I am aware of the fact the audited learning algorithm is different, but this should provide a useful rule of thumb. I suspect this might have to do with the following statement, which might lead to incorrect statistical analysis "We sweep $c' \in \{1,2,...,100\}$ and report the highest $\epsilon$ achieved...". For example, does the analysis take the multiple hypotheses into account?

As mentioned in the previous section, the only novel contribution I have found is Theorem 6 which is essential for performing label-PD auditing under random sampling (that is, no adversarial canaries construction), but considering the poor performance of one run auditing in black box setting even for classical DP, I fail to understand how it can be even remotely useful for label DP, where the label is predicted quite well by the features, leaving very little randomness to begin with.

**Minor comments**
1. I fail to understand the need for the simulation based privacy definition. It seems like this is akin to the commonly used zero-out adjacency notion for the label-DP setting.
2. I found the choice to denote $D^{b}$in Algorithm 2 as $o_{2}$ despite representing a very different quantity to $o_{1}$ quite confusing and I recommend avoiding it.
3. The choice of $\tau$ in Theorem 5 is confusing, since $\tau$ represents the TV distance in Theorem 6 and this quantity is denoted by $\gamma$ in the algorithm statement.
4. More generally, Theorem 5 is nearly impossible to understand without reading the abstract, as it is stated in terms of Algorithm 3 which does not appear in the main body.

**Questions:**

Please address the concerns raised in the previous section.

---

> ### Author Response · Authors · 2025-11-21
>
> Thank you for your careful reading and thoughtful review. We first address the weaknesses.
>
> **How does the observational game differ from the setting where $2n$ points are sampled?**
> > From the problem setting perspective, I do not understand how does the supposedly new observational auditing method differ from a simple setting where $2n$ points are sampled from $\mathcal{D}$, they are matched in pairs, from which one is added to the training set as random. [...]
>
> - Theoretically, the attack you describe with $2n$ datapoints matched in pairs, the attack where $n$ additional datapoints are sampled after training finishes, and the attack in Steinke et al where we flip a coin for each datapoint, are all the same. We explain this in the paragraph “An observational versus an interventional membership inference attack” (lines 214-226). We also explain that all of these variants were known prior to our work, although without giving much attention to whether they are observational.
> - These differences matter in practical applications. If one is just given access to the final trained model and cannot re-train it (e.g., it is too expensive to re-train or we do not have access to the model), the attack with $2n$ datapoints and the attack of Steinke et al are no longer feasible.
> - Our most important contribution is that the $n$ additional datapoints need not be sampled from $\mathcal{D}$, which would again be difficult in a production setting where you may not have access to or ability to affect the training pipeline. The alternative is to generate these datapoints synthentically, which itself is notoriously challenging. However, and this is our key observation, synthetically generating a proxy _label_ is easier than synthetically generating an entire sample or a subset of the attributes.
>
> **Theoretical results, in particular is Theorem 5 simply a special case of Steinke et al.**
> >On the theoretical level, the contribution is not clear to me. The Label DP aspect cannot affect the privacy guarantees, since the public features are akin to any other auxiliary information publicly known to the mechanism and auditor alike. The restriction to canaries sampled from the underlying distribution rather than adversarial constructed only weakens the auditing power, so I fail to understand what does what is the purpose of Theorem 5 if it is simply a special case of the known result by Steinke et al.
>
> - Theorem 5 is not too different in its proof from Steinke et al, there is however a key subtle difference regarding observational vs interventional attacks.
> - As written, Steinke et al’s proof relies on the interventional ability to affect the training set. In their proof, changing the random bits flipped by the challenger directly affects the training data and thus the mechanism’s output, which allows for the differential privacy guarantee to be applied.
> - We write this proof in a somewhat more general form, where intervention is not necessary for the proof to go through: by changing the value of the random bit b in Algorithm 2, we do not affect the training of M, but we affect the label received by the adversary, which in turn affects the adversary’s output, which in turn depends on M. The adversary’s output is a post-processing of M, and we apply the differential privacy guarantee to the adversary’s output rather than to the outputs of M.
>
> Theorem 6 is our key contribution, with Theorem 5 stated as a stepping stone to Theorem 6.

---

> > ### Author Response · Authors · 2025-11-21
> >
> > **Experimental results on CIFAR-10**
> > > On the experimental front I find the results surprising and somewhat suspicious [...]
> >
> > A key point here is that our attack is _not_ one-run. Steinke et al’23 use the one-run attack known as “loss attack” (https://arxiv.org/pdf/1709.01604) in their experiments, which indeed has been shown to be very weak by multiple works, e.g., Watson et al. In contrast, the Observational LIA attacker has access to the reference model M’ that it uses to calibrate its scores (Lines 344-345). In the MIA setting, having access to just one \emph{additional} reference model gives much stronger attacks than the loss attack, as shown in Watson et al, which we compare to, and in the “Low-Cost High-Power Membership Inference Attacks” paper (Robust MIA, https://arxiv.org/pdf/2312.03262).
> >
> > Some additional points:
> > - While our attack is not one-run as it relies on the additional model M’, in lines 254-262 we explain that obtaining M’ need not require additional training. Rather M’ can be an earlier checkpoint of the model M that we target.
> > - We do not account for multiple hypothesis testing, which indeed introduces statistical bias, although the hypotheses here are highly correlated. The same approach has been used in Steinke et al (see page 25, https://arxiv.org/pdf/2305.08846). We can also show results for the adversary that always guesses on 1% of samples (the highest epsilon results are usually achieved at this value).
> > - Our experimental results use f-DP auditing which produces tighter auditing than the method of Steinke et al (e.g., Figure 2 in https://arxiv.org/pdf/2410.22235 shows nearly tight epsilon lower bound at eps=1, albeit for a white-box attack).
> >
> >
> > **Next steps**
> > To recap, the actionable steps for us based on this review are
> > - Clarify that our attack is not one-run, to avoid confusion when comparing to results in Steinke et al
> > - Clarify that we do not account for multiple hypothesis testing and show results for an attacker that guesses on only 1% of samples.
> > - Provide more explanation for how Theorem 5 differs from prior work.
> >
> >
> > **Minor comments:**
> > - Need for simulation-based DP definition: zero-out is somewhat awkward to state for label-DP as it would require arbitrarily replacing labels by 0. A bottom-out (replacing the label with a special symbol) is equivalent to the simulation-based definition; we prefer introducing the simulator as the more explicit way of stating that the new label can be treated differently. (Importantly, the simulator knows what label needs to be protected.).
> > - Use of $o_2$and $o_1$ in Algorithm 2: we will remove these
> > - Using $\tau$ in Theorem 5 is an oversight, it should be replaced by $\gamma$.
> > - With additional space, we can move Algorithm 3 to the main body, although it is not our contribution.

---

> > > ### Comment · Reviewer_jL12 · 2025-11-21
> > >
> > > I thank the authors for their detailed response.
> > >
> > > If I understand correctly, from the theoretical perspective this work should be observed as a sequence of several generalizations and modifications to the one-run-auditing method, which---combined---open the door to a novel auditing method from the practical perspective. Did I get it right?
> > >
> > > If so, I really didn't understand this is the intent of the authors from my first reading, which---while probably partially resulting from lack of care on my side---may confuse other readers. If I may offer a potential solution, I think the theoretical part will greatly benefit from a presentation of its results as a sequence of extensions: label DP which justifies the simulation based neighboring notion, sampling new points as a theoretically identical and practically different variant; usage of samples from a different distribution, etc. Then, the motivational-practical part can put together all these pieces to propose a practical and useful novel auditing process. Given such a presentation it will be easy to disentangle the contribution of each change to the experimental results, hopefully by presenting the effect of implementing only part of the changes (e.g., running the same experiment with fresh elements, rather than fresh labels).
> > >
> > > I will raise my score, but I still think this work will benefit from a significant revision, that will provide future reviewers the ability to get a clearer picture of its contribution, which unfortunately I missed.

---

### Official Review · Reviewer_9U5d · 2025-10-30

**Soundness:** 3
**Presentation:** 3
**Contribution:** 3
**Rating:** 8
**Confidence:** 4

**Summary:**

This paper proposes an observational approach to auditing the privacy parameters of a mechanism under the context of Simulation-based Attribute privacy.
Most prior work on auditing is *interventional* in the sense that the audit needs to change something about the data the mechanism runs on (for example, it might insert canaries) in order to carry out the audit.
This has non-trivial costs for two reasons: first, we care about the mechanism's output on the original data, so if the intervention significantly changes the output of the mechanism then we are paying in utility. Second, there is sometimes non-trivial engineering overhead to modify the mechanism in the ways required by the audit.
By contrast, the audit proposed in this work is *observational* in the sense that the data input to the mechanism is unchanged.

At a high level, the idea behind the proposed observational attack is as follows: Let $D^0$ denote the real dataset which consists of records $(x,y)$ where $x$ is public data that need not be protected and $y$ is sensitive data. Generate a new dataset $D^1$ that contains the same public records $x$, but which resamples the private data for each $x$ from the conditional distribution of $y$ given $x$. Finally, create a mixed dataset $D^b$ by flipping a coin $b_i$ for each row and if $b_i = 0$ we include the row from $D^0$ and if $b_i = 1$ we include the row from $D^1$. Then the attacker observes the output of the mechanism run on $D^0$ and the dataset $D^b$ and is required to guess for each row of $D^b$ whether it came from $D^0$ or $D^1$.
Intuitively, since the rows of $D^0$ and $D^1$ are identically distributed, the only way for the attacker to identify the rows in $D^b$ that come from $D^0$ are to use the fact that the mechanism output was computed from $D^0$.

The authors prove that if an attacker is able to guess a significant number of rows then this leads to a lower bound on the privacy parameters of the mechanism $M$. They also handle the case when $D^1$ contains private data that is resampled from an approximation to the real conditional distribution of $y$ given $x$, in which case the lower bound on $\epsilon$ degrades with the total variation distance between the approximate and true conditional distributions.

Finally, they conduct experiments on CIFAR-10 and Criteo with a concrete attacker and show that the lower bounds on $\epsilon$ computed through their audit are competitive with other auditing techniques for a range of mechanisms.

**Strengths:**

Auditing the privacy parameters of a mechanism without intervention is an important and extremely relevant problem. For example, training or fine-tuning LLMs via DP-SGD may be too expensive to tolerate multiple runs or interventions in the training data that degrade model quality. The proposed auditing setup is reasonable and I believe the lower bounds on the privacy parameters derived from the audit are both believable and useful.

**Weaknesses:**

I have a few minor concerns with the paper that I do not feel should be barriers to its publication, but I do think some additional discussion could be helpful.

First, a subtle point of the audit is that after the attacker commits to their attack $A$, the true data $D^0$ is sampled i.i.d. from the data distribution $\mathcal{D}$.
This randomness appears to be necessary for the analysis to go through, since it is the only way we prevent the attacker $A$ from knowing the records in $D^0$ as side information.
But, in practice, the attacker may legitimately have side-information about $D^0$, especially if it is a benchmark dataset.

For example, in the CIFAR-10 data, we could implement an attacker that accurately guesses the artifacts $b_0, \ldots, b_m$ without looking at the output of the mechanism.
Instead, for each $x_i$ in $D^b$, the attacker searches the public CIFAR-10 data for an exact match and then predicts $b_i = 0$ if $y^{b_i}_i$ matches the label in the public data and predicts $b_i = 1$ otherwise.
This attack will correctly guess $b_i$ for every record such that $y^1_i \neq y^0_i$ (which intuitively seem like the only records where there is any hope to correctly guess better than chance).
But, the success of this attack should not provide a lower bound on the privacy parameters of the mechanism $M$, since it did not involve $M$ at all.
Practically speaking, I think this just means that we need to be careful when we design the attack for the audit: we should try as much as possible to not let the design of the attacker be influenced by the true dataset.

The second concern is that this audit will only provide non-trivial lower bounds on the privacy parameters in situations where the conditional distribution of $y$ given $x$ is not too concentrated. For example, $y$ is a deterministic function of $x$, then when we resample $y^1_i$ from the conditional distribution given $x_i$, we will always have $y^1_i = y^0_i$ and so $D^1 = D^0$ and it seems that this should imply that no attacker can do better than chance. This is a minor concern because in these situations, the guarantee provided by attribute privacy is already weak (e.g., if your private record is a deterministic function of your public record, then there is nothing you can do to hide it whether the mechanism $M$ operates on your data or not).

**Questions:**

I would be especially curious to hear your thoughts about the requirement that $D^0$ be random.

A related question is whether or not it is important that $D^0$ contain i.i.d. records drawn from some distribution $\mathcal{D}$ (as opposed to a more complicated data collection process)

---

> ### Author Response · Authors · 2025-11-21
>
> Thank you for the careful reading and your thoughtful review.
>
> Your observation about using publicly known data in the attack is very interesting. I think iid sampling is not a necessary condition for theoretical results to go through, but it is needed for Theorem 6 to give meaningful privacy auditing.
>
> To elaborate, we state our Theorem 6 in terms of the TV distance between $D’|x$ and $D|x$ (taking the maximum over all $x$ in the training set). However, it can also be stated in terms of the adversary’s ability to distinguish between $y\sim D|x$ and $y’\sim D’|x$ given any limits on side information or resources that we want to place on the adversary. If $(x, y)$ is public, then $y$ and $y’$ are very distinguishable as you say – but then Theorem 6 says we cannot extract much privacy leakage this way. This leads to your remark that it is the responsibility of the attack designer to account for the information the attacker may possess.
>
> Regarding your comment on how $D|x$ needs to have enough entropy – that is indeed the case, the entropy is what makes the observational attack possible. We do not recommend using this attack in cases where the label can be easily derived from the features, e.g., in images (even though we run CIFAR-10 experiments as it is a popular dataset), but rather in cases where the label represents a person's preferences, actions, beliefs.

---

### Official Review · Reviewer_Agft · 2025-10-31

**Soundness:** 2
**Presentation:** 4
**Contribution:** 3
**Rating:** 4
**Confidence:** 4

**Summary:**

This submission proposes an observational privacy auditing framework: instead of modifying the training data or pipeline (no canaries, no deletions), it leverages randomness in the data distribution itself to audit privacy for protected attributes such as label-DP, and gives a way to convert attack success rates into lower bounds on (ε,δ). The key assumption is the existence of a proxy label distribution that is (for a reasonable attacker) indistinguishable from the true one; in streaming/incremental training, an earlier checkpoint can serve as this proxy, keeping engineering cost very low. They cast auditing as a challenger–adversary game, define SIM-DP via an “imputation simulator,” and show empirically that their observational audit yields ε lower bounds comparable to calibrated, lightweight MIA baselines on CIFAR-10 and Criteo.

**Strengths:**

The presentation quality is pretty good and the story is easy to follow.

The problem is indeed an important one because the efficiency has always been the bottleneck of auditing DP in practice.

The theory makes sense and the derived method show good performance.

**Weaknesses:**

There is not obvious weakness of theory and intuition part. But some weaknesses in the experiments part may downgrade of the soundness of the contributions you claimed.

Have you ever tried to audit the membership-inference-target DP, e.g., the traditional DP-SGD on bounding indistinguishability of each sample? How does it look like? Although the attribute inference is an important problem, the traditional membership inference is also important. Now that you show your game based on sample-level membership inference problem, I hope to see how your architecture perform in practice.

The methods are only tested on two relatively simple set. You claimed that the observational method makes auditing efficient and can be deployed in practice, e.g., some larger models or datasets. However, the experiments are only on two datasets. Criteo is fine, but CIFAR-10 is way too simple. At least you should try your method in some cases where previous methods are not feasible, e.g., huge dataset or model.

Although the method looks efficient, but no one knows what will happen in practice. You should mention the speed up of your methods in experiments to support the claim that your observational way does accelerate auditing. Or you can simply do the amortized analysis on your method and previous baseline methods.


I think the paper's intuition is novel and the problem itself is important. But on these three weaknesses of evaluation, I cannot say the soundness is good. However, if you can show some more evaluation to support your claim "\textit{By lowering the complexity of privacy auditing, our approach enables its application in a wider variety of contexts.}" and solve these weaknesses, I'll be happy to increase my scores.

**Questions:**

Please see weaknesses.

---

> ### Author Response · Authors · 2025-11-21
>
> Thank you for the careful reading and your valuable input. We’ll address some of the weaknesses.
>
> **Auditing DP-SGD or other mechanisms from the MIA literature**
> > Have you ever tried to audit the membership-inference-target DP, e.g., the traditional DP-SGD on bounding indistinguishability of each sample? How does it look like? Although the attribute inference is an important problem, the traditional membership inference is also important. Now that you show your game based on sample-level membership inference problem, I hope to see how your architecture perform in practice
>
> We did not attempt auditing mechanisms that are not explicitly label-DP as in these cases we do not know the ground truth epsilon label-DP guarantee (we only know the full-sample-DP upper bound), and thus do not have a baseline against which to compare our empirical epsilon lower bound. Our attack can be used with such mechanisms and we expect that it would show a similar empirical epsilon as MIAs that train only one additional model. We believe such exploration, i.e., understanding the power of label inference attacks compared to MIAs, is better suited for follow-up work as it would require more extensive exploration and ideally some theoretical results.
>
> **Experimenting with larger datasets**
> > The methods are only tested on two relatively simple set. You claimed that the observational method makes auditing efficient and can be deployed in practice, e.g., some larger models or datasets. However, the experiments are only on two datasets. Criteo is fine, but CIFAR-10 is way too simple. At least you should try your method in some cases where previous methods are not feasible, e.g., huge dataset or model
>
> Thank you for this suggestions. We have experimented with datasets that are common in the literature on privacy auditing, but we can also add experiments using a larger text dataset for a sentiment analysis task.
>
> **Efficiency of the method**
> >Although the method looks efficient, but no one knows what will happen in practice. You should mention the speed up of your methods in experiments to support the claim that your observational way does accelerate auditing. Or you can simply do the amortized analysis on your method and previous baseline methods.
>
>
> Thank you for raising this point as we believe it merits further explanation in our work. When running experiments for the paper, the LIA takes the same amount of time to run as the calibration-based MIA of Watson et al: both methods require training the target model and one additional reference model. Both attacks run linearly in the number of canaries.
>
> Where our attack distinguishes itself from other attacks is in a production setting. Let’s say we have trained a large model with several checkpoints and we would like to audit this model post hoc. This is impossible with MIA but can be done without any modifications to the model for LIA. To clarify, to run an MIA we need to have access to a dataset that is guaranteed to not be part of training and has the same distribution as the training data. Finding such datasets has been a big challenge in auditing LLMs, but can also be difficult for recommender models where one usually consumes all training data available and tests on subsequent days’ data, which has a different distribution. To prepare a canary dataset for an MIA, it is necessary to exclude some of the training data ahead of time. In row 1, table 1, the cost of model-retraining is the dominant cost.
>
> **Table 1**. Complexity of auditing for MIA vs LIA, where m is the number of canaries and k is the cost of forward pass for model inference.
>
>
>
> |Setting | Cost of LIA | Cost of MIA |
> |----------|:--------:|---------:|
> |Auditing an already trained model | $O(km)$ | Cost of LIA + cost of model re-training|
> |Online training|$O(km)$ per checkpoint|Cost of LIA + cost of maintaining a training pipeline that consistently sets aside data to be used as canaries|
>
>
>
> Now further consider a setting where we would like to run the attack repeatedly for each new model checkpoint deployed in production. Row 2 compares the costs. The cost of maintaining data pipelines is hard to quantify as it involves human effort, but is clearly an additional significant cost compared to running LIA.
>
> Please let us know if such comparison of costs sufficiently illustrates the complexity gains of our method.
>
> **Next steps**
>
> To recap, based on this review we will
> - Add experiments with one more larger text dataset
> - Provide a longer discussion of the efficiency gains with LIA

---

> > ### Comment · Reviewer_Agft · 2025-11-27
> >
> > Thanks for your answer. I will keep my score because I think the first question, i.e., how it perform on DP-SGD, is vital to me.  label-DP is a relatively narrow topic in privacy, I think discussing only on the label-DP limits the paper's contribution.

---

### Official Review · Reviewer_w4LJ · 2025-11-03

**Soundness:** 3
**Presentation:** 4
**Contribution:** 4
**Rating:** 8
**Confidence:** 4

**Summary:**

This paper proposes a novel approach for privacy auditing of machine learning models, which relies on the intrinsic characteristics of the data distribution. The main strength of the proposed approach is that it does not require to modify the training dataset. However, the approach works in the restricted setting of label differential privacy.

**Strengths:**

-The paper is well-written but an outline at the end of the introduction would help to clarify its structure. The privacy auditing is also well-explained and formally proven and is based on previous results from the auditing literature.

-One of the novel aspect of the proposed auditing approach is that it relies on an attribute inference attack rather than a membership inference attack. Additionally, it does not require an alteration of the training pipeline of the model.

-The approach has been validated experimentally on two datasets and against a wide range of mechanisms ensuring label-DP. The results demonstrate that the approach is able to obtain meaningful estimate for low value of epsilon.

**Weaknesses:**

-The setting considered is that of label DP, which is rather restricted as it aims to protect only the privacy of the label of a particular record. Thus, while the approach is interesting and has the benefit of not require to alter the training pipeline, its applicability beyond label DP mechanisms appear to be limited.

-The number of canaries reported for the Criteo and CIFAR-10 experiments is quite high. Ideally, the paper should contain some figures displaying how the quality of the estimate is impacted by the number of canaries. Furthermore the quality of the estimate appears to be quite low on Criteo for values of epsilon beyond 2.

-A key component of the approach is the proxy model that is trained to infer a label based from the profile itself. The paper is currently lacking an in-depth discussion on the possible alternative to create it and the impact that it has on the quality of the estimate.

-The comparison with other state-of-the-art approaches for privacy auditing is limited as it consists only to the MIA approach proposed by Watson et al. 2022.

A small typo :
-« and the of the trained » -> « and of the trained »

**Questions:**

Please see the main points raised in the weaknesses section.

---

> ### Author Response · Authors · 2025-11-21
>
> Thank you for the careful reading and your valuable input. We’ll address some of the weaknesses.
>
> **Restricted applications of Label-DP**
> > The setting considered is that of label DP, which is rather restricted as it aims to protect only the privacy of the label of a particular record. Thus, while the approach is interesting and has the benefit of not require to alter the training pipeline, its applicability beyond label DP mechanisms appear to be limited.
>
> There are multiple settings where the label is the most sensitive and valuable piece of information, as it encodes user preferences and actions (e.g., likes, views, clicks, purchases). Ranking or recommender systems, such as ads or content personalization, are the principal domain of label privacy. In addition, auditing label-DP implies lower bounds for traditional DP, thus our attack could be used for privacy auditing beyond the setting of label-DP mechanisms.
>
> **Impact of the number of canaries**
> > The number of canaries reported for the Criteo and CIFAR-10 experiments is quite high. Ideally, the paper should contain some figures displaying how the quality of the estimate is impacted by the number of canaries.
>
> One of the benefits of our observational attack is that one can use as many canaries as the size of the training set. The more canaries we use, the narrower the confidence interval for the empirical epsilon estimate. This is different from MIA where the number of canaries presents a tradeoff: more canaries means one needs to reduce the size of the training data whereas fewer canaries lead to wider confidence intervals (discussed in Steinke et al’23). For a fair comparison, we use the same number of canaries for LIA as MIA, which is the entire size of the test set.
>
> **Quality of epsilon estimates for values above 2**
> > Furthermore the quality of the estimate appears to be quite low on Criteo for values of epsilon beyond 2.
>
> The gap between theoretical and empirical epsilons especially for black box attacks in the high-epsilon regime is a commonly observed phenomenon (e.g., Figure 8, black box setting in Steinke et al’23 https://arxiv.org/pdf/2305.08846, where theoretical epsilon=8, empirical epsilon=2). Moreover, a tight epsilon guarantee for the label-DP algorithms is not known and would be hard to estimate, thus we only know an upper bound.
>
> **Discussion of the proxy model**
> >A key component of the approach is the proxy model that is trained to infer a label based from the profile itself. The paper is currently lacking an in-depth discussion on the possible alternative to create it and the impact that it has on the quality of the estimate.
>
>
> In lines 254-261 (“Obtaining approximate distributions”) we discuss that obtaining the proxy model is a key aspect of the attack. In online machine learning systems with recurring training one can use prior model checkpoints to generate the labels for running the attack on the latest checkpoint. While our theorems take into account the gap in the quality of the proxy model (via the TV distance), observing these effects experimentally may indeed be valuable. We will extend our experiments on synthetic data to thta end.
>
>
> **Comparison to state-of-the-art**
> > The comparison with other state-of-the-art approaches for privacy auditing is limited as it consists only to the MIA approach proposed by Watson et al. 2022.
>
> Since the motivation for our method is to obtain attacks that require minimal overhead relative to the standard ML training, we limited our comparison to methods that likewise require fewer resources, in particular attacks with no more than 1 additional shadow/reference run. Watson et al is one such method though we could potentially also compare to Robust MIA (https://arxiv.org/abs/2312.03262) with 1 shadow run. We expect that attacks with several shadow runs such as LiRA (https://arxiv.org/abs/2112.03570) would give better results than our LIA, which is intended as a lightweight attack.
>
> **Next steps**
>
> To recap, we will take the following actionable items based on your suggestions
> - Expand discussion and provide experiments on synthetic data that evaluate the role of the quality of the proxy model.
> - Compare to one additional attack: RMIA (https://arxiv.org/abs/2312.03262)
> - Add a clear list of contributions at the end of the introduction

---

### Author Response · Authors · 2025-12-03

Thank you to the reviewers for the many useful suggestions and the rebutal engagement, which has greatly helped improve our paper. We upload a revised version of the manuscript that addresses reveiwer comments.

Our changes include:
- Additional experiments with 1 larger language dataset Twitter sentiment analysis, (showing similar results as our our prior experiments on 2 datasets)
- Additonal experimental comparison to one more attack from the prior literature (showing similar results as prior comparsion)
- Additional experiments with synthetic data that demonstrate the effect of distribution shift in our theorem
- Further discussion in the Introduction about our contributions, including an itemized list of contributions
- Further clarifications throughout the main body of the paper on points raised by reviewers

We provide a summary of the state of the rebuttal up to November 27
- Reviewer w4LJ:     inital score 8
- Reviewer Agft:       inital score 4, commented on Nov 27 that they would maintain their score
- Reviewer 9U5d:     inital score 8
- Reviewr jL12:         inital score 2, commented on Nov 21 that they would increase their score upon seeing revised mansucript
- Reviewer 5SJ9:      inital score 2, commented on Nov 21 that they would incrase their score to 6 upon seeing revised manuscript

---

### Note · Program_Chairs · 2026-01-17
**Submission Desk Rejected by Program Chairs**

The following references in this submission do not refer to real documents and/or have major errors in bibliographic information:

 1. Jack Watson, Matthew Jagielski, Florian Tramèr, and Nicholas Carlini. Difficulty calibration and membership inference attacks. In International Conference on Learning Representations (ICLR), 2022.

2. Congzheng Song and Vitaly Shmatikov. Membership inference attacks against collaborative filtering algorithms. In 2019 IEEE Symposium on Security and Privacy (S&P), pp. 457-474. IEEE, 201